# Oxytocin administration enhances pleasantness and neural responses to gentle stroking but not moderate pressure social touch by increasing peripheral concentrations

Yuanshu Chen[1], Haochen Zou[1], Xin Hou[2], Chuimei Lan[1], Jing Wang[3], Yanan Qing[1], Wangjun Chen[1], Shuxia Yao[1]*, Keith M Kendrick[1]*

[1]The Clinical Hospital of Chengdu Brain Science Institute, MOE Key Laboratory for NeuroInformation, Center for Information in Medicine, University of Electronic Science and Technology of China, Chengdu, China; [2]School of Educational Sciences, Chongqing Normal University, Chongqing, China; [3]West China School of Pharmacy, Sichuan University, Chengdu, China

*For correspondence:
yaoshuxia@uestc.edu.cn (SY);
k.kendrick.uestc@gmail.com
(KMK)

Competing interest: The authors declare that no competing interests exist.

## Abstract

**Background:** Social touch constitutes a key component of human social relationships, although in some conditions with social dysfunction, such as autism, it can be perceived as unpleasant. We have previously shown that intranasal administration of oxytocin facilitates the pleasantness of social touch and activation of brain reward and social processing regions, although it is unclear if it influences responses to gentle stroking touch mediated by cutaneous C-touch fibers or pressure touch mediated by other types of fibers. Additionally, it is unclear whether endogenous oxytocin acts via direct entry into the brain or by increased peripheral blood concentrations.

**Methods:** In a randomized controlled design, we compared effects of intranasal (direct entry into the brain and increased peripheral concentrations) and oral (only peripheral increases) oxytocin on behavioral and neural responses to social touch targeting C-touch (gentle-stroking) or other (medium pressure without stroking) cutaneous receptors.

**Results:** Although both types of touch were perceived as pleasant, intranasal and oral oxytocin equivalently enhanced pleasantness ratings and responses of reward, orbitofrontal cortex, and social processing, superior temporal sulcus, regions only to gentle-stroking not medium pressure touch. Furthermore, increased blood oxytocin concentrations predicted the pleasantness of gentle stroking touch. The specificity of neural effects of oxytocin on C-touch targeted gentle stroking touch were confirmed by time-course extraction and classification analysis.

**Conclusions:** Increased peripheral concentrations of oxytocin primarily modulate its behavioral and neural responses to gentle social touch mediated by C-touch fibers. Findings have potential implications for using oxytocin therapeutically in conditions where social touch is unpleasant.

**Funding:** Key Technological Projects of Guangdong Province grant 2018B030335001.
**Clinical trial number:** NCT05265806

## Editor's evaluation

The therapeutic promise of oxytocin to ameliorate deficiencies in social interactions and/or reward circuitry has been confounded by conflicting literature regarding routes of administration and

regions of impact (i.e. central or peripheral). This important study systematically compares oral versus nasal administration on the pleasantness of gentle stroking, which is c-fiber mediated, and massage, which is multimodal. The convincing results are unambiguous that either route increases perceived pleasantness only to gentle stroking and that the effects, while perceived in the brain, are likely mediated peripherally.

## Introduction

Social touch, is of great importance for social interactions and individual development and can promote interpersonal communication (*Cascio et al., 2019*; *Jones and Glover, 2014*). In conditions where social dysfunction occurs, such as autism, social touch is often perceived as unpleasant (*Baranek et al., 2006*; *Ujiie and Takahashi, 2022*). Touch perception is determined by the stimulation of low-threshold afferent fibers in the skin that innervate distinct classes of mechanoreceptors. While large myelinated Aβ fibers are densely packed in the fingertips and lips and subserve discriminative touch, unmyelinated C-touch (CT) fibers exist in hairy skin and only respond to low force/velocity (caress-like stroking) touch (*Croy et al., 2016*; *McGlone et al., 2014*). The latter fibers are specialized for the affective domain of touch and have evolved to signal the rewarding value of physical contact (*Liljencrantz and Olausson, 2014*; *Pawling et al., 2017*; *Walker et al., 2017*). Stimulation of the CT system primarily engages the insula via the spinothalamic tract and brain circuits involved in reward and social-emotional information processing such as the orbitofrontal cortex (OFC) and posterior superior temporal sulcus (pSTS) (*Björnsdotter et al., 2014*; *Davidovic et al., 2016*; *Gordon et al., 2013*; *Morrison, 2016*; *Olausson et al., 2002*; *Olausson et al., 2010*). Medium pressure touch in the form of hugging or massage can also be perceived as pleasant but mainly influences pressure receptors of non-CT fibers (*Case et al., 2021*; *Field, 2010*) and primarily targets the somatosensory cortex via the spinothalamic tract (*McGlone et al., 2014*). While neural substrates of pleasurable gentle stroking and medium pressure touch overlap to some extent they may also involve different parts of the somatosensory cortex and insula (*Case et al., 2021*).

The neuropeptide oxytocin (OT) plays a key role in the regulation of social cognition and the rewarding aspects of social stimuli, including social affective touch (*Bartz et al., 2011*; *Kendrick et al., 2017*; *Rae et al., 2022*; *Wigton et al., 2015*). There is a close association between OT and CT-targeted social touch (*Moberg et al., 2020*; *Uvnäs-Moberg et al., 2014*; *Walker et al., 2017*) and studies have shown that social touch, particularly administered as gentle stroking touch or medium pressure massage, can activate parvocellular oxytocinergic neurons (*Okabe et al., 2015*; *Tang et al., 2020*) and facilitate endogenous OT release in the saliva, blood or urine across species (*Crockford et al., 2013*; *Holt-Lunstad et al., 2008*; *Li et al., 2019*; *Morhenn et al., 2012*; *Portnova et al., 2020*; *Vittner et al., 2018*). Additionally, intranasal administration of OT in humans can modulate processing of social touch at both behavioral and neural levels. More specifically, intranasal OT enhances the perceived pleasantness and activity of the reward system and salience network in response to gentle social touch by a female in men (*Scheele et al., 2014*), by their partners rather than by an unfamiliar female (*Kreuder et al., 2017*), or during foot massage administered by a human but not by a machine (*Chen et al., 2020b*). Intranasal OT also enhances the pleasantness of gentle stroking touch administered indirectly via different materials independent of valence (*Chen et al., 2020a*). These findings consistently suggest that intranasal OT can enhance the hedonic properties of social touch. While previous studies have interpreted findings in terms of OT potentiating the rewarding effect of CT-targeted social touch, it is less clear whether it does the same for pleasant stimulation of non-CT fiber mechanoreceptors which can also be stimulated by social behaviors such as hugging and during medium pressure massage. Foot massage, for example, incorporates both gentle stroking and medium pressure massage and thus the observed release of OT (*Li et al., 2019*) and facilitatory effects of intranasal OT (*Chen et al., 2020b*) may be contributed by CT and non-CT fibers. There is some evidence that receiving frequent hugs can increase OT concentrations (*Grewen et al., 2005*; *Light et al., 2005*), although hugging is often accompanied by gentle stroking of the back or neck and so once again both CT and non-CT fibers may be involved. It therefore remains unclear which types of cutaneous afferent fibers are primarily involved in the functional effects of OT on social touch processing.

The mechanism(s) whereby intranasal OT produces its functional effects are currently a matter of debate (see *Leng and Ludwig, 2016*; *Yao and Kendrick, 2022*). A number of studies have established that intranasally administered OT can directly enter the brain via the olfactory and trigeminal nerves (see *Lee et al., 2020*; *Quintana et al., 2021*) and it has been widely assumed to be the main route whereby intranasal OT produces effects on brain and behavior. However, intranasally administered OT also enters the peripheral circulation after absorption by nasal and oral blood vessels and may produce functional effects by entering the brain after binding to the receptor for advanced glycation end products (RAGE) and crossing the blood brain barrier (BBB; *Yamamoto and Higashida, 2020*), or by vagal stimulation following stimulation of its receptors in the heart and gastrointestinal system (see *Carter, 2014*; *Carter et al., 2020*; *Yao and Kendrick, 2022*) or following stimulation of receptors in other organs. Indeed, in a recent study we have shown that reducing the entry of oxytocin into the peripheral circulation following intranasal administration using a vasoconstrictor prevents its effects on resting state electroencephalographic changes involving cross-frequency coupling (*Yao et al., 2023*). In the specific context of responses to tactile stimulation peripheral OT could influence OT receptors in keratinocytes in the skin which may act to modulate activity of responses of cutaneous sensory fibers (*Baumbauer et al., 2015*; *Deing et al., 2013*; *Talagas and Misery, 2019*) or in spinal dorsal route ganglion neurons which receive inputs from cutaneous sensory fibers and project to the brain via the spinothalamic tract (*González-Hernández et al., 2017*; *Noguri et al., 2022*).

Support for peripherally mediated routes have been found in animal model studies reporting functional effects of OT administered subcutaneously or intraperitoneally (see *Yao and Kendrick, 2022*), and in a previous study, we have shown similar effects of intranasal and oral (lingual) OT on visual attention and state anxiety (*Zhuang et al., 2022*). There may however also be some administration route-dependent functional effects of OT. For example, in monkeys intranasal and intravenous administration of OT have been reported to produce different patterns of regional perfusion (*Lee et al., 2018*), although in humans they produced similar neural effects (*Martins et al., 2020b*). Another recent study in humans has shown that intranasal and oral (lingual) OT have different effects on amygdala and putamen responses to emotional faces and on associated arousal (*Kou et al., 2021*). Thus, it is unclear whether the reported effects of exogenously administered OT on responses to social touch are mediated via direct entry into the brain or indirectly via increased concentrations in the peripheral circulation which subsequently influence the brain either by crossing the BBB or by acting on receptors in peripheral organs or nerves.

Against this background, the present study investigated firstly whether intranasal OT primarily enhanced the pleasantness and associated brain reward responses to touch exclusively targeting CT fibers (using gentle stroking touch) or primarily targeting non-CT fiber pressure mechanoreceptors (medium pressure massage without stroking). Secondly, the effects of intranasal OT were contrasted with those of oral (lingual) OT using the same dose of 24 IU to help establish whether OT was mainly acting via peripherally mediated routes (see *Figure 1A* for a complete study procedure). While intranasally administered OT can produce functional effects by either directly entering the brain or by increasing peripheral vascular system concentrations, oral OT administration can only do so by the latter. We therefore postulated that if equivalent effects of the two routes of administration are observed then the functional effects of OT are likely to be due to peripheral vascular increases. Behavioral responses to different types of touch were recorded using rating scales and neural responses were acquired using functional near-infrared spectroscopy (fNIRS) measures of oxygenated-hemoglobin chromophore concentration changes in cortical brain regions involved in reward (OFC), social cognition (STS) and somatosensory processing (somatosensory cortex - S1; *Figure 1B–C*). fNIRS is now widely used in neuroimaging (*Pinti et al., 2020*) and has the advantage that subjects are more comfortable and can move more than in an MRI scanner and can more easily be catheterized for taking blood samples during recordings. Although, fNIRS only measures activity changes in the superficial cortex it allows recordings to be made from three of the main touch processing regions, the OFC, STS and somatosensory cortex and has therefore often been used in studies investigating touch processing (*Bennett et al., 2014*; *Cruciani et al., 2021*; *Li et al., 2019*). Additionally, physiological measures of autonomic nervous system changes were recorded (skin conductance response - SCR and the electrocardiogram - ECG) and blood samples were taken for measurement of OT concentration changes. We hypothesized firstly that while subjects would find both the gentle stroking touch and medium pressure massage pleasurable, intranasal OT would particularly enhance behavioral and brain

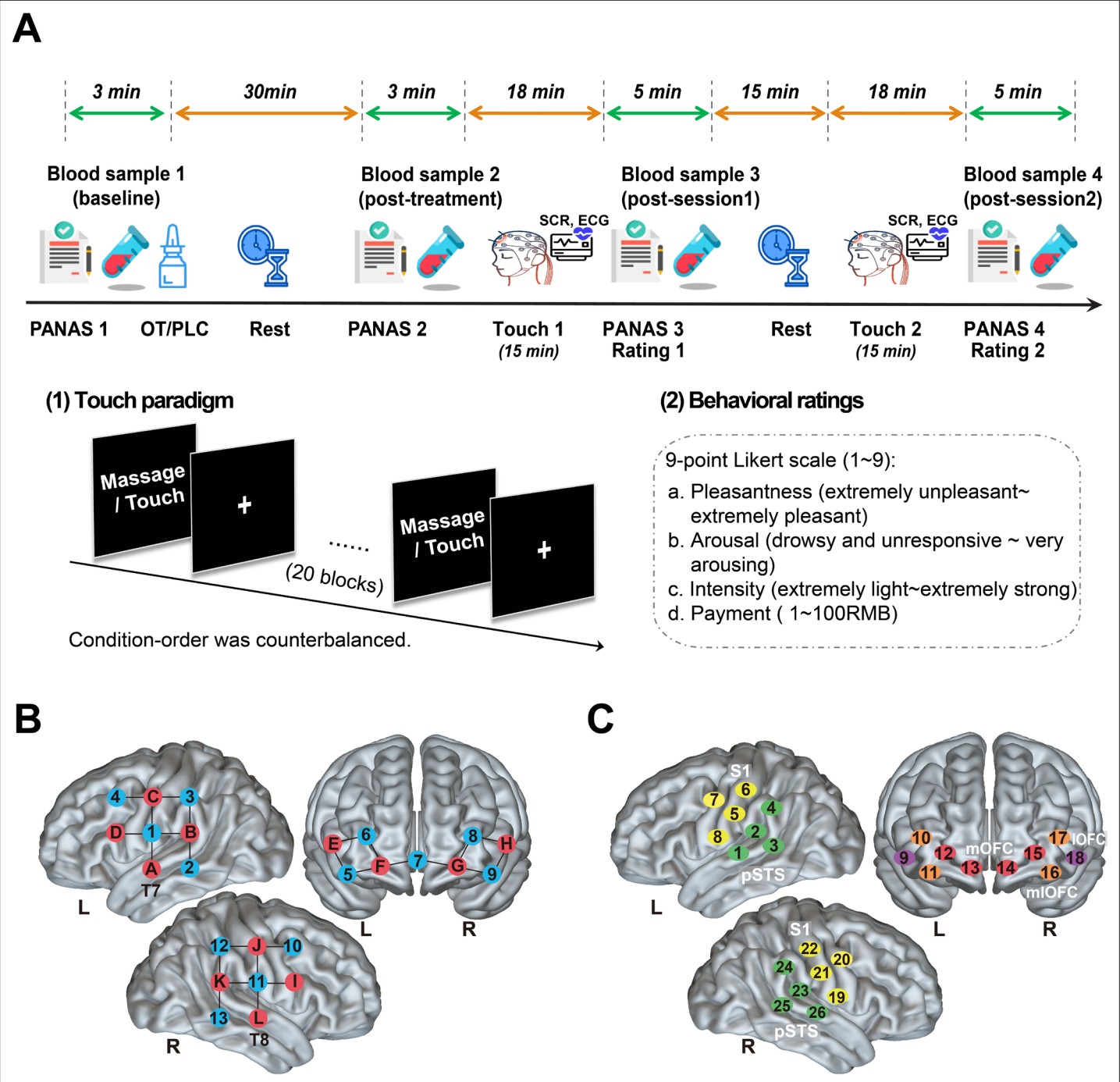

**Figure 1.** The study procedure and layout of fNIRS optodes and channels. (**A**) The study protocol and sequences of the experimental task. A total of four blood samples (6 ml for each) were collected for each subject before and after the treatment and after each session to measure OT concentration changes. Subjects completed the positive and negative affective schedule (PANAS) before and after the intranasal/oral OT or PLC treatment. During each touch session, neural responses were acquired using functional near-infrared spectroscopy (fNIRS) measures and physiological measures of autonomic nervous system changes including the skin conductance response - SCR and the electrocardiogram - ECG were recorded as well. Subjects were subsequently asked to rate their mood (PANAS) and subjective experience of the massage/touch including the perceived pleasantness, arousal, intensity, and willingness to payment after each session. (**B**) The array design displayed the locations of the sources (red) and detectors (blue). (**C**) Channels according to the international 10–20 placement system. A 26-channel array consisting of 12 sources and 13 detectors were used to record hemodynamic activity of the bilateral lateral orbitofrontal cortex (lOFC: channels 9, 18), medial orbitofrontal cortex (mOFC: channels 12–15) and mediolateral orbitofrontal cortex (mlOFC: channels 10, 11, 16, 17), posterior superior temporal sulcus (pSTS: channels 1–4, 23–26) and the somatosensory area (S1: channels 5–8, 19–22).

*Figure 1 continued on next page*

*Figure 1 continued*

The online version of this article includes the following figure supplement(s) for figure 1:

**Figure supplement 1.** CONSORT flow diagram.

reward (OFC) and social processing (STS) region responses to gentle stroking touch targeting the CT fiber system in line with our previous studies (*Chen et al., 2020a*; *Scheele et al., 2014*) and not in response to the medium pressure massage targeting non-CT fibers. Secondly, based on recent studies comparing effects of intranasal and oral OT (*Xu et al., 2022*; *Zhuang et al., 2022*), we hypothesized that oral OT would have a similar effect to intranasal OT, thereby indicating that its effects were mediated by increased concentrations in the peripheral vasculature rather than by direct entry into the brain. Thirdly, we hypothesized that behavioral and neural effects of OT on social touch would be associated with treatment effects on peripheral plasma concentrations of the peptide, in line with observations in some previous studies (*Martins et al., 2020b*; *Kou et al., 2021*).

## Methods

### Participants

An a priori sample size calculation using G* power indicated that 159 participants (number of groups: 3, and 53 subjects per group) should be sufficient to reliably detect a medium effect size ($\alpha$=0.05, $f$=0.25, 80% power). A total of 180 healthy Chinese subjects (90 males, 21.22±2.77 years) participated in the present study. Exclusion criteria consisted of any self-reported psychiatric/physical illness, alcohol/substance abuse, or other major health concern. Self-reported menstruating information and the luteinizing hormone test were conducted for all female participants. They were asked to participate in the experiment avoiding the menstruation periods. All subjects gave written informed consent prior to any study procedures. All experimental procedures were in accordance with the latest revision of the declaration of Helsinki and approved by the local ethics committee of the University of Electronic Science and Technology of China and registered as a clinical trial (NCT05265806). Five subjects were excluded due to failure to complete the procedures and four subjects were dropped because of technical problems during data acquisition (details see *Figure 1—figure supplement 1*). Thus data from a final sample of 171 subjects (87 females, 21.58±1.94 years) were analyzed (intranasal OT: N=56; oral OT: N=57; PLC: N=58) (see *Table 1*).

### Blood samples and OT assay

To measure plasma OT concentrations, before and 30 min after intranasal and oral (lingual) treatments, as well as immediately after each session of social touch/massage condition (see *Figure 1A* for sampling protocol), 4 blood samples in total (6 ml for each) were collected into EDTA tubes from all subjects by an indwelling venous catheter and were stored after centrifugation at –80°C until assays were performed. Oxytocin concentrations were measured using a commercial ELISA (ENZO, USA, kit no: ADI-901–153). Blood samples were analyzed in duplicate and a standard prior extraction step was performed following the recommended protocol from the manufacturers. Spiked samples (with 100 pg/ml OT added) were included with every assay to calculate extraction efficiency which was 96%. The extraction step incorporated a fourfold concentration of samples using a vacuum concentrator (Concentrator plus, Eppendorf, Germany) resulting in a detection sensitivity of 2 pg/ml. All samples had detectable concentrations. The inter- and intra-assay coefficients of variation were 14 and 11% respectively. The manufacturer's reported cross-reactivity of the antibody for other neuropeptides, such as vasopressin and vasotocin, is <0.01% (for detailed OT assay methods, see *Li et al., 2019*).

### Experimental procedure

Participants were first asked to complete Chinese versions of validated questionnaires on personality, traits, mood, attitude toward interpersonal touch and sensitivity to reward to control for possible group differences in potential confounders. Personality trait measures included the Autism Spectrum Quotient (ASQ; *Baron-Cohen et al., 2001*), the Beck Depression Inventory II (BDI; *Beck et al., 1996*), the State-Trait Anxiety Inventory (STAI; *Spielberger et al., 1983*), the Cheek and Buss Shyness Scale (CBSS; *Cheek and Buss, 1981*), the Interpersonal Reactivity Index (C-IRI; *Siu and Shek, 2005*), the

**Table 1.** Demographic, physiological, and psychometric assessments in the three groups (M±SD).

| | Intranasal OT | Oral OT | PLC | *value* | p |
|---|---|---|---|---|---|
| Number (males) | n=56(26) | n=57(27) | n=58(31) | $\chi^2 = 0.67$ | 0.721 |
| Age | 21.79±1.96 | 21.46±2.05 | 21.50±1.84 | F=0.48 | 0.618 |
| ASQ | 21.43±5.48 | 20.09±5.05 | 21.66±5.84 | F=1.37 | 0.256 |
| STQ | 40.36±8.51 | 41.32±9.60 | 40.79±7.67 | F=0.18 | 0.840 |
| SP | 13.16±5.28 | 12.86±5.50 | 13.38±5.48 | F=1.13 | 0.876 |
| SR | 13.20±4.07 | 14.26±4.22 | 14.16±3.60 | F=1.24 | 0.293 |
| SOR | 22.66±9.43 | 20.88±10.23 | 23.17±9.64 | F=0.87 | 0.421 |
| BDI | 9.21±9.11 | 8.39±7.36 | 9.79±8.39 | F=0.42 | 0.660 |
| TAI | 42.96±9.32 | 43.47±9.10 | 43.21±9.77 | F=0.04 | 0.959 |
| SAI | 38.59±9.61 | 38.51±8.37 | 39.02±10.33 | F=0.05 | 0.953 |
| CBSS | 39.46±9.56 | 36.21±9.38 | 38.28±8.98 | F=1.77 | 0.173 |
| CTQ | 38.50±8.57 | 36.51±9.03 | 37.97±9.10 | F=0.76 | 0.469 |
| IRI | 48.89±10.99 | 48.95±10.26 | 50.36±11.09 | F=0.34 | 0.710 |
| AAS | 57.84±6.23 | 58.11±6.04 | 57.26±6.02 | F=0.29 | 0.748 |
| PANAS | | | | | |
| Positive affect | 18.71±0.80 | 19.63±0.78 | 17.42±0.78 | F=2.04 | 0.132 |
| Negative affect | 11.00±0.37 | 11.31±0.36 | 11.15±0.36 | F=0.17 | 0.840 |
| HF | | | | | |
| Gentle stroking touch | 47.24±16.62 | 49.32±17.29 | 49.48±16.47 | | |
| Medium pressure massage | 51.89±17.18 | 53.68±16.20 | 52.89±15.74 | Fa = 0.40 | 0.878 |
| DFAα1 | | | | | |
| Gentle stroking touch | 0.92±0.25 | 0.89±0.24 | 0.88±0.23 | | |
| Medium pressure massage | 0.99±0.25 | 0.94±0.27 | 0.95±0.25 | Fa = 0.33 | 0.922 |
| Heart rate | | | | | |
| Gentle stroking touch | 72.78±8.25 | 74.47±10.45 | 72.47±9.96 | | |
| Medium pressure massage | 71.73±9.56 | 72.09±8.09 | 70.67±9.48 | Fa = 0.70 | 0.653 |
| SCR | | | | | |
| Gentle stroking touch | 1.45±1.81 | 1.20±1.47 | 1.09±1.52 | | |
| Medium pressure massage | 2.48±2.15 | 2.58±2.39 | 2.49±187 | Fa = 0.43 | 0.862 |
| Basal OT concentrations | 8.12±0.54 | 8.49±0.60 | 9.71±0.48 | F=2.39 | 0.105 |

a: F values of the group x condition interaction analyses.

Childhood Trauma Questionnaire (CTQ; *Bernstein et al., 2003*), and the Adult Attachment Scale (AAS; *Collins and Read, 1990*). Individual attitudes and sensitivity to touch and reward were assessed using the Social Touch Questionnaire (STQ; *Wilhelm et al., 2001*), the Sensitivity to Punishment and Sensitivity to Reward Questionnaire (SPSRQ; *Torrubia et al., 2001*), and individual levels of sensory over-responsivity (also referred to as defensive responses) toward tactile stimuli were measured by the Sensory Over-Responsivity (SensOR) Scales (*Schoen et al., 2008*). Additionally, to control for potential confounding effects of treatment on mood, participants completed the Positive and Negative Affect Schedule (PANAS; *Watson et al., 1988*) immediately before and 30 min after the treatment and after each touch session (details see *Figure 1A*).

Following completion of the questionnaire assessments, participants were randomly assigned to receive the oral (lingual) or intranasal administration of OT spray (24 IU; Oxytocin Spray, Sichuan Defeng Pharmaceutical Co. Ltd, China) or their corresponding placebo (PLC) spray (half received the PLC intranasally and half orally; identical ingredients except the peptide - i.e. glycerine and sodium chloride) in a randomized double-blind placebo-controlled between-subject design. Intranasal and oral spray bottles and OT concentrations per 0.1 ml puff were identical (i.e. 4 IU). For intranasal administration, three puffs were applied to each nostril alternating between them and with 30 s between each puff, for oral administration six puffs were sprayed onto the tongue (i.e. lingual) with 30 s between each puff and the subjects required not to swallow until just before the next puff was applied to allow time for absorption by oral blood vessels (as in **Kou et al., 2021**). Subjects and experimenters were blind concerning whether PLC or OT was administered. Blinding for the two different routes of administration was not possible although subjects were not informed until they arrived to sign the consent form and take part in the experiment whether they would receive intranasal or oral treatment. In line with our previous findings that oral and intranasal PLC administration do not produce different functional effects either at neural or behavioral levels (**Kou et al., 2021**; **Zhuang et al., 2022**), there were also no significant differences in pleasantness ratings of the gentle stroking touch and medium pressure massage between the intranasal and oral PLC groups (*p*s >0.110) confirming that knowledge of the route of administration had no effects. We therefore combined them into a single PLC group. Participants were unable to guess better than chance whether they had received OT or PLC ($\chi^2$=0.43, p=0.512). As a further control for the lack of blinding in the groups receiving different routes of administration initial analyses of blood samples, behavioral and neural data were performed blind by experimenters.

In line with previous studies (**Paloyelis et al., 2016**; **Spengler et al., 2017**), the task started approximately 35–40 min after the treatment administration. Neural and physiological responses to social touch stimulation were measured via simultaneously acquired fNIRS together with SCR and ECG recording. A professional masseur blinded to the research aim was trained by the experimenter to administer the different social touch stimuli to the calf of each leg as consistently as possible. For gentle touch, the masseur applied only a light stroking touch at the optimum velocity for activating CT fibers (5 cms/s) and at which subjects perceive this form of touch as most pleasant (**Löken et al., 2009**). For the massage stimulation condition, the masseur applied a medium pressure massage moving discretely up and down the leg at the same velocity but without stroking the skin, designed to primarily target non-CT fibers (**Field, 2010**). The gentle touch and medium pressure massage were delivered on both legs simultaneously to control for possible preferences for left or right and more importantly to avoid unilateral brain activation. During the experiment, the masseur could simultaneously see a visual cue indicating the type of stimulation on a personal monitor and was instructed to vary the exact start and end point of each stimulation on the calf randomly by a few millimeters in order to minimize receptor fatigue (**Cascio et al., 2012**). To reduce the possibility that subjects might be uncomfortable with receiving gentle stroking touch from a stranger, subjects were informed that both types of touch stimulation they would receive were forms of professional massage using different amounts of pressure and that they just needed to be relaxed and concentrate on how the administered 'massage' made them feel. They were further informed that either a masseur or a masseuse would be randomly assigned by the experimenter to deliver the 'massage', although in fact, they were always given by a same masseur. The paradigm consisted of two sessions and each session comprised 20 blocks of gentle stroking touch or medium pressure massage. Condition-order was counterbalanced across participants. Each block lasted for 30 s alternated with a rest interval of 15 s and each session lasted for 15 min with a 15 min rest interval between each (**Figure 1A**).

Immediately after each session, subjects completed the PANAS and then answered the following four questions: (1) How pleasant did you feel the massage? (1=extremely unpleasant, 9=extremely pleasant). (2) How much did the massage arouse you? (1=drowsy and unresponsive, 9=very arousing). (3) How intense was the massage? (1=extremely light, 9=extremely strong) and (4) How much would you be willing to pay if you had to pay for the applied massage? Please choose from 1 to 100 (1=1 RMB, 9=100 RMB). After the experiment, participants were asked to guess the gender of the massager to control for possible sex-dependent effects.

## Physiological data acquisition and analyses

Physiological measures were collected at a sampling rate of 1000 Hz using a Biopac MP150 system (Biopac Systems, Inc) and recorded using AcqKnowledge (Version 4.4, Biopac Systems Inc, CA, USA). SCR was recorded using a GSR100C module with two electrodes being placed on the tips of participant's left index and middle fingers. ECG was recorded using an ECG100C module with three electrodes (including the ground electrode) placed relatively close to each other in parallel on the left side of the upper torso (see *Niendorf et al., 2012*).

Physiological data were analyzed using the AcqKnowledge 4.4 software following the manual. To determine different SCR amplitudes in response to the medium pressure massage and the gentle stroking touch, we computed the mean base-to-peak difference within a 15 s time window after the stimulation and rest onset. SCR differences were compared across the three treatment groups for each stimulation condition and rest as well as between the medium pressure massage and the gentle stroking touch. The raw ECG data was band-pass filtered (range: 0.5–35 Hz; 8000 coefficients) to remove baseline drift and high-frequency noise. A template correlation function was used to transform noisy data manually. Next, we extracted R-R intervals from the clean ECG which were next imported into Kubios software (http://kubios.uku.fi) for heart rate (as a sympathetic nervous system measure of arousal) and HRV (indexed by the high-frequency component — HF and the detrended fluctuation scaling exponent — DFAα1, assumed to reflect parasympathetic influence) analyses (*Kemp et al., 2012*; *Martins et al., 2020a*).

## fNIRS data acquisition and analyses

Hemodynamic response signals were acquired using the NIRSport2 System (NIRx Medical technologies LLC, Berlin, Germany) operating at two wavelengths (760 and 850 nm) with a sampling frequency of 6.78 Hz. In line with previous studies (*Li et al., 2019*; *Long et al., 2021*; *Tsuzuki and Dan, 2014*), each optode was placed on the surface of skull according to reference points on the head (the nasion, inion, left and right ears, top and back of the head) adjusting for different head size and shapes of different participants. The probe set contains 12 sources and 13 detectors with 3 cm source-detector separation to cover brain regions of interest (ROIs) and allows for 26 different channels to measure the oxyhemoglobin and deoxyhemoglobin concentration changes. The current study focused on the oxyhemoglobin (oxy-Hb) concentration changes because of the higher sensitivity to cerebral blood flow changes and better signal-to-noise ratio. Based on previous studies (e.g., *Li et al., 2019*; *Bennett et al., 2014*), five regions engaged in touch processing were selected as a priori ROIs, including the bilateral lateral OFC (lOFC), medial OFC (mOFC), mediolateral OFC (mlOFC), posterior STS (pSTS) and primary somatosensory cortex (medial S1). The optodes were placed in accordance with the international 10–20 system and the lowest lines of the probes were placed at T7 and T8 corresponding to positions of channels 1 and 25, respectively (see *Figure 1B–C*).

The fNIRS raw data were analyzed using the NIRS-KIT software which can be used for both resting-state and task-based fNIRS data analyses (*Hou et al., 2021*). During preprocessing, raw optical intensity data were firstly converted to concentration changes of oxy-Hb based on the modified Beer-Lambert law. A polynomial regression model was then applied to remove linear detrends from the raw time course. Motion-related artifacts and baseline shifts were removed using the temporal derivative distribution repair (TDDR) method and an infinite impulse response (IIR) Butterworth bandpass filter (0.01–0.08 Hz) was additionally employed to correct for machinery and physiological noise. On the first level, a generalized linear model (GLM) approach was applied to model the task-related hemodynamic response with four regressors (the medium pressure massage, the gentle stroking touch, the rest after the massage and the rest after the touch) in the design matrix. Contrast images for each stimulation condition minus the rest were created for each participant receiving intranasal/oral OT or PLC treatments. On the second level, beta estimates obtained from each channel of each participant were extracted and analyzed using SPSS 26 (IBM, Inc).

We also employed the representational similarity analysis (RSA) technique to investigate whether the received treatment of one participant (test subject, leave-one-out method) could be correctly classified based on their similarity of activation patterns to the mean pattern of subjects receiving the same treatment (group models except for the test subject each time; *Emberson et al., 2017*; *Hernandez et al., 2018*). Mean time course of the mOFC, mlOFC and pSTS within the time window at 5–35 s following the gentle touch stimulation onset was extracted from each participant as input

features. The number of correctly labeled subjects were averaged to compute the actual accuracy for discrimination between the two treatments (intranasal OT vs. PLC, oral OT vs. PLC, intranasal OT vs. oral OT; scripts adapted from *Emberson et al., 2017*; details see *Figure 5—figure supplement 1*). Permutation-based significance tests via creating a null distribution of treatment labels (permutations = 10000) were conducted to investigate whether the actual accuracies based on correct treatment labels significantly differed from those computed from randomly assigned treatment labels. p Values for the actual classification accuracies were computed by measuring what proportion of observed accuracies on the null distribution are equal to or greater than the actual classification accuracy. Decoding accuracy was also computed for discrimination between the two stimulation conditions (see *Figure 5—figure supplement 1*).

## Statistical analyses

For plasma OT concentration analyses, firstly one-way ANOVA for mean basal concentrations of plasma OT was conducted to compare group differences. Secondly, one-way ANOVA was applied for post-treatment OT concentration changes (minus the basal OT concentrations) to investigate whether OT administration significantly increased plasma OT concentrations compared with PLC. Finally, treatment effects on changes of plasma OT concentrations after the gentle stroking touch versus the medium pressure massage were examined by a repeated-measures ANOVA with treatment (intranasal OT vs. oral OT vs. PLC) as between-subject factor and stimulation condition (medium pressure massage vs. gentle stroking touch) as within-subject factor. Effects on behavioral rating scores for pleasantness, arousal, intensity and payment willingness and physiological indices (HRV and SCR amplitudes) were respectively examined using repeated-measures ANOVAs with treatment as between-subject factor and stimulation condition as within-subject factor. A repeated-measures ANOVA was employed to examine OT's effects on massage and touch with treatment as between-subject factor and stimulation condition and brain region (lOFC vs. mOFC vs. mlOFC vs. pSTS vs. S1) as within-subject factors. We also performed channel-by-channel repeated-measures ANOVAs to confirm the ROI-based analysis results (see *Figure 3—figure supplement 1*). All post-hoc analyses were applied with Bonferroni correction to disentangle significant main effects and interactions. Effect size was calculated using the partial eta-squared and two tailed *p* values were reported. Correlations between OT-induced behavioral/neural effects and blood OT concentration changes were calculated using Spearman correlation coefficients. All statistical analyses were performed using SPSS version 22.

## Results

### Demographics and questionnaire scores

Repeated-measures analyses of variance (ANOVAs) with treatment as between-subject factor revealed no significant differences between treatment groups (*p*s >0.130) with respect to demographics and questionnaire scores measuring personality traits and mood (*Table 1* and *Supplementary file 1* Table 1A).

### The effects of OT and social touch on plasma OT concentration changes

A one-way ANOVA showed no between-group difference in basal plasma OT concentrations taken prior to the treatment (F(2, 168)=2.39, p=0.105; *Table 1*). A one-way ANOVA on increases in plasma concentrations relative to basal levels revealed a significant main effect of treatment (F(2, 168)=29.46, p<0.001) with higher plasma OT concentration changes at 30 min (before the task) following intranasal (mean increase = 6.26 ± 0.93 pg/ml, p<0.001) and oral OT treatment (mean increase = 2.10 ± 0.42 pg/ml, p=0.016) compared with PLC (mean increase = –0.31 ± 0.35 pg/ml). Plasma OT concentrations following intranasal treatment were also significantly higher than following oral treatment (p<0.001, *Figure 2A* and *Supplementary file 1* Table 1B).

To investigate changes of plasma OT concentrations after the gentle stroking touch relative to the medium pressure massage, a repeated-measures ANOVA on plasma OT concentrations measured after each touch stimulation session with treatment as between-subject factor and condition as within-subject factor revealed a significant main effect of treatment (F(2, 166)=14.20, p<0.001, $\eta_p^2 = 0.15$) indicating that plasma OT concentrations in the intranasal (p<0.001) and oral OT groups (p=0.003) following the gentle stroking touch and medium pressure massage were both significantly higher than in the PLC

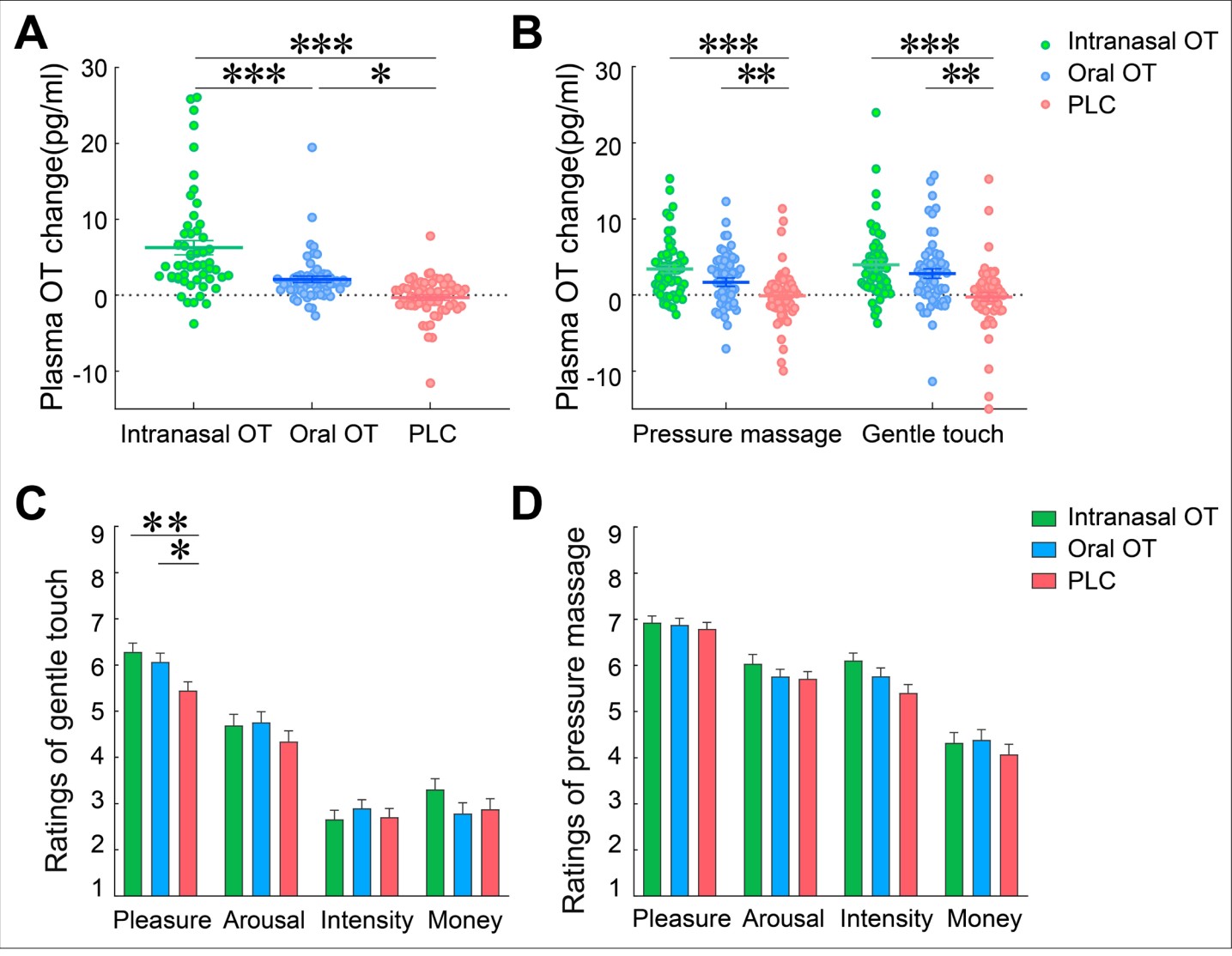

**Figure 2.** Effects of oxytocin on plasma OT concentration changes and behavioral rating scores. (**A**) Post-treatment changes of plasma OT concentrations (compared with pre-treatment baseline). (**B**) Plasma OT concentration changes (compared with pre-treatment baseline) after gentle stroking touch and medium pressure massage following the OT and PLC treatments. Behavioral rating scores of pleasantness, arousal, intensity, and payment willingness following intranasal OT (N = 56), oral OT (N = 57) and PLC (N = 58) treatments in response to (**C**) gentle stroking touch and (**D**) medium pressure massage. One-way ANOVA for post-treatment OT concentration changes (minus the basal OT concentrations) was conducted to compare group differences. Repeated-measures ANOVAs were applied to investigate changes of plasma OT concentrations and treatment effects on behavioral response after the gentle stroking touch versus the medium pressure massage. Error bars show standard errors. $^*p < 0.05$, $^{**}p < 0.01$, $^{**}p < 0.001$ between group comparisons.

group, whereas no difference was found between intranasal and oral OT treatment groups (p=0.162). There was also a marginal main effect of condition (F(1, 166)=3.39, p=0.067, $\eta_p^2 = 0.02$), with plasma OT concentrations being higher following the gentle stroking touch (mean increase: 2.18±0.36 pg/ml) than medium pressure massage (mean increase: 1.67±0.30 pg/ml). The interaction between treatment and stimulation condition was not significant (F(2, 166)=1.89, p=0.154; *Figure 2B*). Overall therefore, both intranasal and oral OT increased plasma OT concentration changes (relative to basal levels) relative to PLC, 30 min after treatment administration and after the two touch sessions, with some indication of levels being higher after gentle stroking touch relative to medium pressure massage.

## The effects of OT on behavioral rating scores and physiological indices

A repeated-measures ANOVA on pleasantness rating scores revealed a significant main effect of treatment (F(2, 168)=3.76, p=0.025, $\eta_p^2 = 0.04$) due to subjects in the OT groups rating both types of social

touch stimulation more pleasant than the PLC group. The main effect of stimulation condition was also significant (F(1, 168)=58.42, p<0.001, $\eta_p^2$ = 0.26), with subjects rating the medium pressure massage (6.87±0.08) more pleasant than the gentle stroking touch (5.95±0.11). Importantly, there was also a significant treatment x condition interaction (F(2, 168)=3.23, p=0.042, $\eta_p^2$ = 0.04). Post-hoc Bonferroni corrected tests showed that both intranasal OT (p=0.006) and oral OT (p=0.044) significantly increased pleasantness ratings of the gentle stroking touch compared with PLC group (*Figure 2C*), but not for the medium pressure massage (ps >0.960) (*Figure 2D*).

Analyses of other ratings revealed a significant effect of stimulation condition with subjects perceiving the medium pressure massage more arousing (F(1, 168)=44.62, p<0.001, $\eta_p^2$ = 0.21), and intense (F(1, 168)=634.83, p<0.001, $\eta_p^2$ = 0.79) and being willing to pay more for it (F(1, 168)=91.38, p<0.001, $\eta_p^2$ = 0.35) than the gentle stroking touch. There were no significant main effects of treatment or interactions for arousal (ps >0.138) or intensity (ps >0.107) ratings, or ratings of willingness to pay (ps >0.354). Additional analyses of possible gender and menstrual cycle-dependent effects revealed no significant effects on pleasantness ratings (gender: ps >0.316, menstrual cycle: ps >0.110). Thus, these factors may not confound treatment effects on pleasantness perception of social touch. Moreover, the perceived gender of the masseur did not significantly influence the pleasantness ratings in either of the stimulation condition (ps >0.441) or on the effects of OT on increasing pleasantness ratings of gentle stroking touch (ps >0.364; see *Supplementary file 1* Table 1C). Thus, both intranasal and oral OT only significantly increased individual perceived pleasantness (not ratings of arousal, intensity and payment) of the gentle stroking touch, but not for the medium pressure massage and this behavioral effect remained stable after controlling for possible confounding effects.

Analyses of SCR data showed a significant main effect of condition (ps <0.001) with greater increases in SCR amplitude following both touch conditions compared with rest as well as following medium pressure massage relative to gentle stroking touch (see *Table 1* and *Supplementary file 1* Table 1D). However, there were no significant treatment effects (ps >0.380) providing additional evidence that OT did not influence levels of arousal in response to either type of touch. Both heart rate and all measures of heart rate variability (HRV) also showed main effects of condition (ps <0.001) with higher HRV during both touch conditions compared with rest and during medium pressure massage relative to gentle stroking touch, and higher heart rate induced by gentle stroking relative to medium pressure massage (see *Table 1* and *Supplementary file 1*, Table 1D). However, once again there were no effects related to treatment (ps >0.490). Thus, both types of touch influenced sympathetic nervous system measures of arousal (increased SCR and heart rate) and parasympathetic measures of vagal tone (increased HRV), although medium pressure was more potent than gentle stroking touch with the exception of heart rate. However, OT treatment did not influence either sympathetic or parasympathetic responses to touch.

## The effects of OT on neural responses

To determine the effects of OT on neural responses to gentle stroking touch and medium pressure massage in the a priori ROIs, a repeated-measures ANOVA for oxy-Hb concentration changes was employed and showed a significant main effect of brain region (F(4, 672)=18.70, p<0.001, $\eta_p^2$ = 0.10) and significant region x treatment (F(8, 672)=1.92, p=0.051, $\eta_p^2$ = 0.02) and region x condition interactions (F(4, 672)=14.41, p<0.001, $\eta_p^2$ = 0.08). Post host comparisons showed a significant oxy-Hb increase in the mOFC following oral OT (p=0.020) and gentle stroking touch induced greater oxy-Hb concentration changes in the mOFC, mlOFC, and pSTS than the medium pressure massage (ps <0.05), whereas in the medial S1 greater changes were induced by the medium pressure massage (p=0.002). Importantly, the three-way condition x region x treatment interaction was also significant (F(8, 672)=2.11, p=0.033, $\eta_p^2$ = 0.03). Post-hoc Bonferroni corrected pair-wise comparisons for each ROI revealed that both intranasal and oral OT significantly enhanced the activity in the bilateral mOFC (intranasal OT: $p_1$=0.029, oral OT: $p_2$=0.005), mlOFC (intranasal OT: $p_1$=0.005, oral OT: $p_2$=0.030) and pSTS (intranasal OT: $p_1$=0.043, oral OT: $p_2$=0.040) compared with the PLC only in response to gentle stroking touch. There were no significant treatment effects on neural responses in the lOFC (ps >0.559) and S1 (ps >0.993). For the medium pressure massage, no significant treatment effects were found (ps >0.237) (*Figure 3*).

Extraction of time courses of oxy-Hb concentration changes showed distinct hemodynamic patterns in response to medium pressure massage and gentle stroking touch. In comparison with

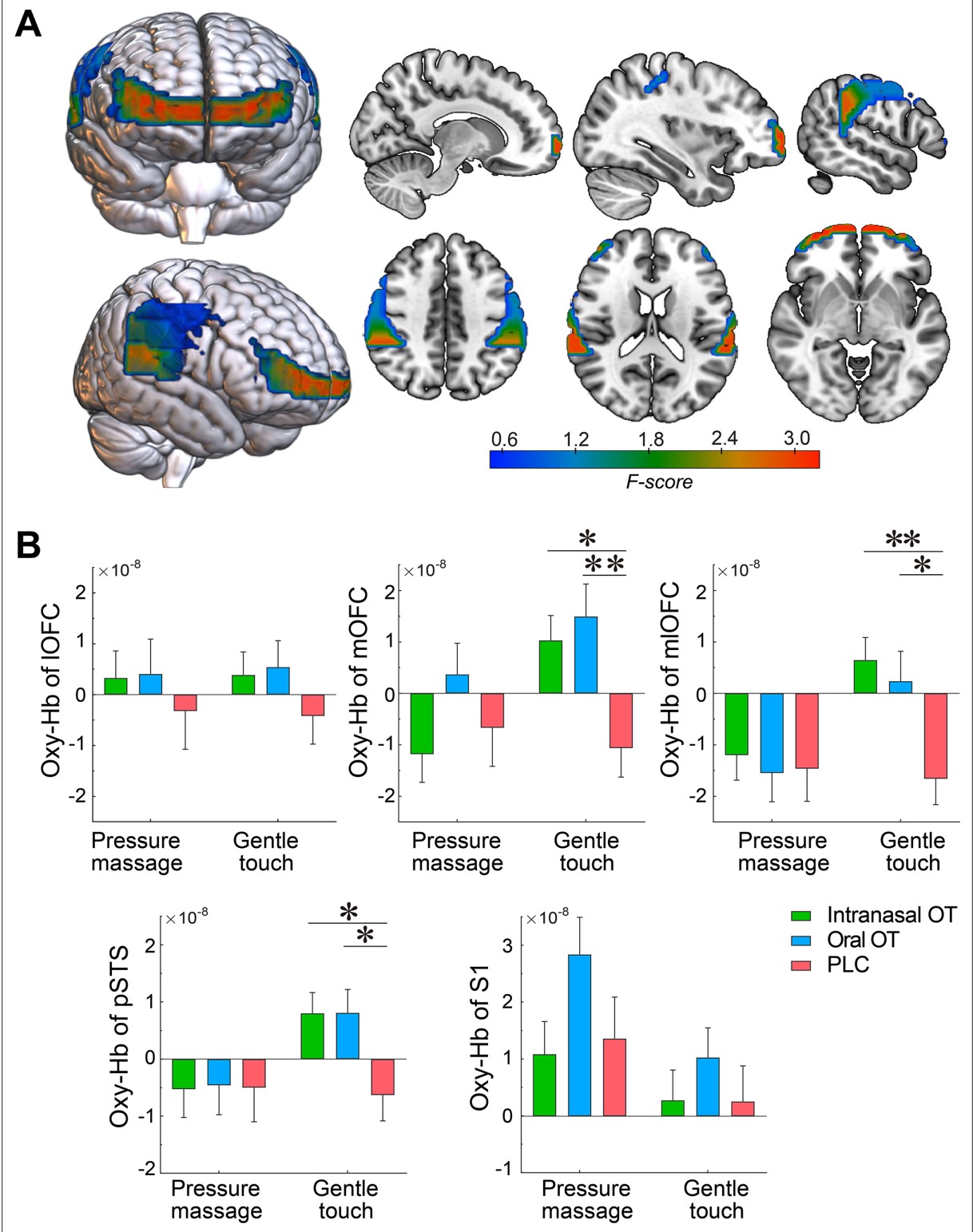

**Figure 3.** Effects of intranasal and oral OT on neural responses to gentle stroking touch. (**A**) Heat brain maps show treatment x condition interaction effects (F values) on neural activations in each ROI using repeated-measures ANOVAs. (**B**) Averaged oxy-Hb concentration changes in the bilateral lOFC, mOFC, mlOFC, pSTS and S1 (mean ± SEM) in response to the gentle stroking touch and medium pressure massage in intranasal OT (N = 56), oral OT (N = 57) and PLC (N = 58) groups. Repeated-measures ANOVAs were applied to investigate treatment effects on neural response to the gentle stroking touch versus the medium pressure massage. $^{*}p < 0.05$, $^{**}p < 0.01$.

*Figure 3 continued on next page*

*Figure 3 continued*

The online version of this article includes the following figure supplement(s) for figure 3:

**Figure supplement 1.** Activation maps of the channel-by-channel analysis.

---

oxy-Hb concentration showing a convergent tendency across intranasal OT, oral OT and PLC groups after offset of the massage stimulation, both intranasal and oral OT prolonged the response in the bilateral mlOFC, mOFC and pSTS to gentle stroking touch (*Figure 4*). We therefore conducted a classification analysis to determine whether time courses of oxy-Hb concentration changes can discriminate different groups. Results showed that time courses of the mOFC, mlOFC and pSTS could achieve a modest level to discriminate between intranasal OT and PLC (mean accuracy = 70.1%, permutation test p<0.001) and between oral OT and PLC groups (mean accuracy = 64.3%, permutation test p=0.008). However, the classification accuracy for discriminating between intranasal and oral OT groups was not significantly higher than the classification accuracy on the null distribution (mean accuracy = 51.3%, permutation test p=0.494; see *Figure 5*). Taken together, results suggest that both intranasally and orally administered OT produced similar functional effects on neural activations in the mOFC, mlOFC and pSTS during CT-fiber-mediated social touch.

### Associations between different outcome measures and mediation analyses

Post-treatment OT concentration changes (30 min post-treatment and before task) were found to be positively correlated with pleasantness rating scores (Spearman's rho = 0.18, p=0.017) and oxy-Hb concentration changes in the mlOFC (Spearman's rho = 0.16, p=0.044) in response to the gentle stroking touch.

Mediation analyses showed that intranasal and oral OT both increased post-treatment plasma OT concentrations (intranasal OT: path *a*=6.57, p<0.001; oral OT: path *a*=2.41, p=0.005) and pleasantness rating scores for gentle stroking touch (intranasal OT: path *c*=0.79, p=0.004; oral OT: path *c*=0.66, p=0.015). The plasma OT concentration changes were found to be positively associated with perceived pleasantness of the gentle stroking touch (intranasal OT: path *b*=0.06, p=0.008; oral OT: path *b*=0.06, p=0.008) and when included as a mediator in the model, results although showed no significant direct effects of intranasal and oral OT on the perceived pleasantness (intranasal OT: path *c'*=0.37, p=0.228; oral OT: path *c'*=0.50, p=0.062). By contrast, the results showed that post-treatment plasma OT concentration changes totally mediated the enhancement effects of intranasal (indirect effect = 0.42, SE = 0.166, 95% CI = [0.101, 0.768], bootstrap = 5000, *Figure 6A*) and oral OT (indirect effect = 0.15, SE = 0.071, 95% CI = [0.037, 0.310], bootstrap = 5000, *Figure 6B*) on pleasantness ratings. Similar mediation analyses of post-treatment plasma OT concentration changes and the OFC/pSTS activity in response to gentle stroking touch revealed no significant results (*p*s >0.103). Results from the model suggest that behavioral effects of both intranasally and orally administered OT were primarily mediated by peripheral changes in plasma OT concentrations following treatments.

## Discussion

The current study aimed firstly to establish whether effects of OT treatment on enhancing the perceived pleasantness of social touch were primarily in response to CT fiber stimulation or that of non-CT fibers. Our findings that OT only facilitated behavioral and neural responses to gentle stroking touch support the view that it is primarily influencing CT fiber mediated touch. The second aim of the study was to establish whether the route of exogenous OT administration is important. Here findings that similar functional effects of OT were produced after intranasal and oral administration, and that post-treatment changes in plasma OT concentrations significantly mediated behavioral effects, support the conclusion that the observed effects were primarily mediated by peripheral changes.

Gentle stroking touch and medium pressure massage were both perceived as pleasant although the medium pressure massage was rated more pleasant, stronger and more arousing and subjects were more willing to pay for it, possibly because the latter is closer to the familiar experience of normal massage. Medium pressure massage also produced greater increases in autonomic measures of arousal (SCR) and vagal tone (HRV), although gentle stroking touch increased heart rate slightly

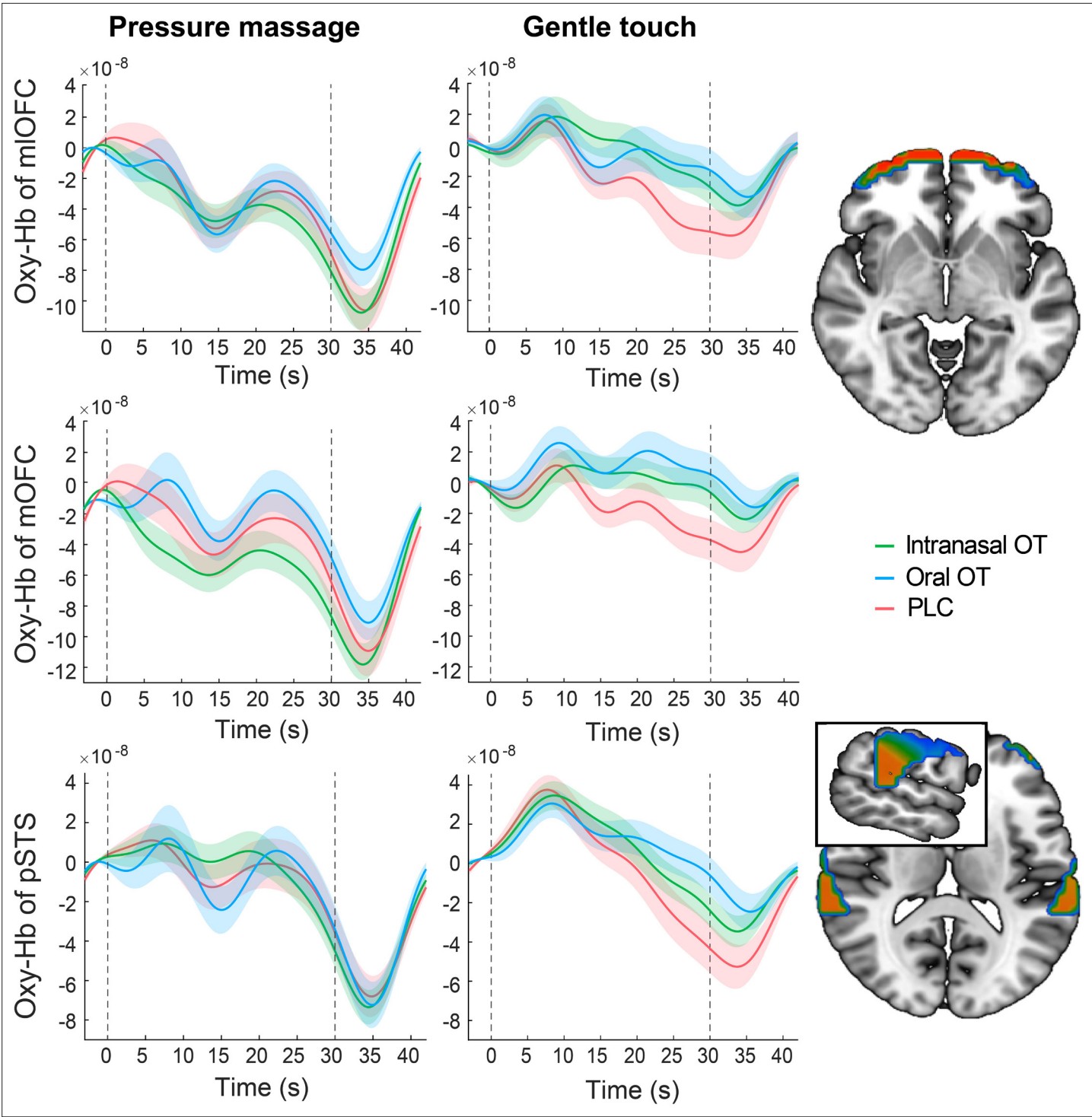

**Figure 4.** Time courses of oxy-Hb concentration changes in response to medium pressure massage and gentle stroking touch in the bilateral mlOFC, mOFC and pSTS for intranasal OT (N = 56), oral OT (N = 57) and PLC (N = 58) groups, respectively. The dotted gray lines indicate the onset and offset of the stimulation and the shaded areas represent ± SEM.

more. Additionally, the medium pressure massage produced greater oxy-Hb concentration changes in the S1 region, in line with the greater perceived stimulation intensity. However, both intranasal and oral OT selectively increased pleasantness ratings for the gentle stroking touch but not medium pressure massage. Consistent with our previous study (*Chen et al., 2020b*), there were no effects of OT on ratings of intensity, arousal, or willingness to pay, suggesting a specific modulation of hedonic

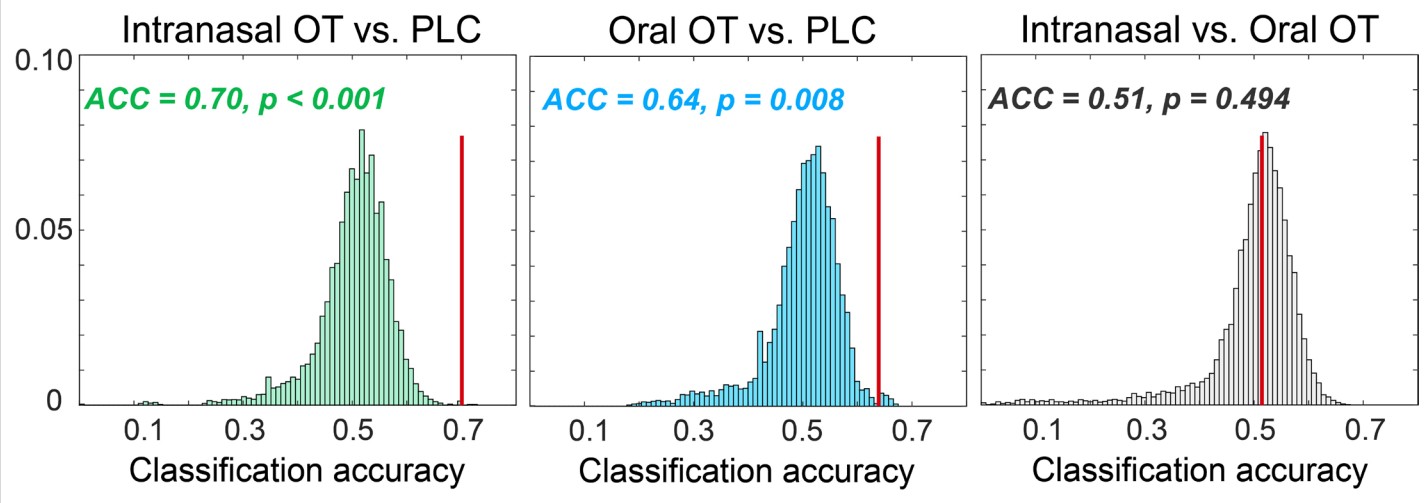

**Figure 5.** Null distribution for classification accuracies for discriminating intranasal OT versus PLC, oral OT versus PLC, and intranasal versus oral OT groups, respectively. Histograms show the distribution of accuracies from permutation tests (permutations = 10,000). The red lines indicate the actual accuracy and p values for the actual classification accuracies.

The online version of this article includes the following figure supplement(s) for figure 5:

**Figure supplement 1.** Illustration of the multivariate analysis method.

pleasantness, and subjects were unable to guess if the touch/massage was administered by a male or female. Thus, in support of previous suggestions (*Walker et al., 2017*) OT may be selectively increasing the hedonic pleasure of touch primarily stimulating CT fibers rather than non-CT fiber pressure mechanoreceptors. Furthermore, the effects of both intranasal and oral OT on pleasantness ratings following the gentle stroking touch were correlated with, and mediated by, post-treatment increases in plasma OT concentrations.

Consistent with previous studies showing enhanced brain reward effects of intranasal OT during tactile stimulation (e.g. *Chen et al., 2020a*; *Scheele et al., 2014*), the present study found that both intranasal and oral administration of OT enhanced responses in the bilateral mlOFC during gentle stroking touch. Although subjects rated medium pressure massage as more pleasant it is possible

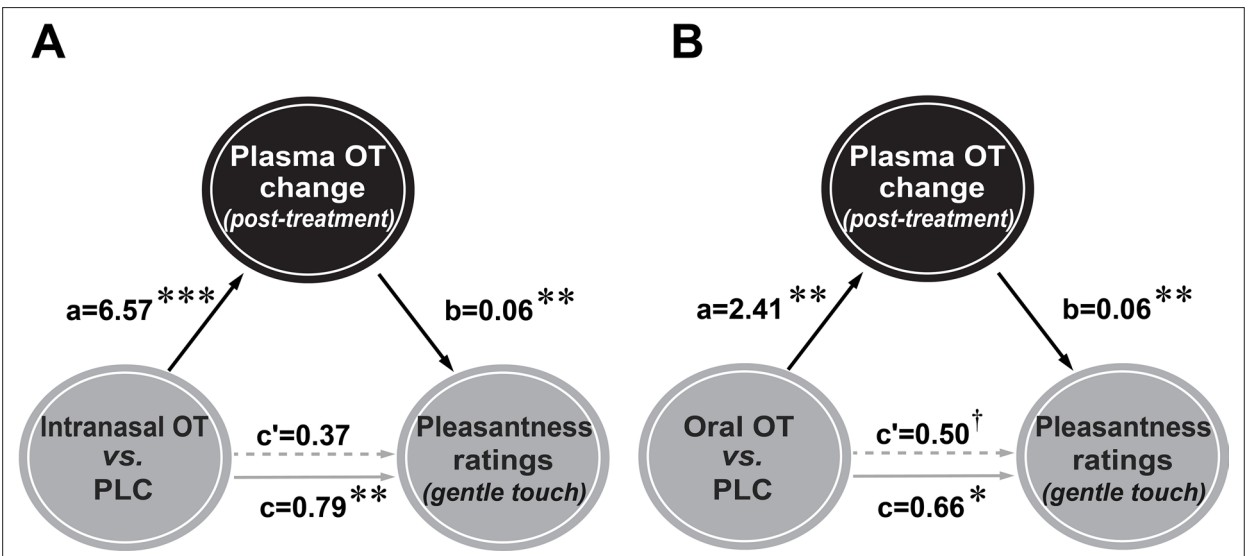

**Figure 6.** Mediation analyses for intranasal and oral oxytocin (OT) effects on pleasantness ratings via plasma OT concentrations. Enhanced pleasantness ratings in response to gentle stroking touch following intranasal (**A**) and oral (**B**) OT administration were significantly mediated by post-treatment plasma OT concentration changes. [†]$p=0.05$, [*]$p < 0.05$, [**]$p < 0.01$, [***]$p < 0.001$.

that the greater responses to gentle stroking touch in the mOFC and mlOFC may reflect not only reward but also interactions with brain regions such as the cingulate cortex and insula which process social salience (see *Rolls et al., 2020*). Furthermore, both routes of OT administration enhanced pSTS responses to gentle stroking touch. This region is important for both touch and social cognition processing and pSTS responses are predictive of the perceived pleasantness of caress-like skin stroking (*Davidovic et al., 2016*). The specificity of pSTS responses to social touch is supported by previous observations that tactile stimuli using materials or by some kind of mechanical device did not report activations in this region (*Boehme et al., 2019*; *Chen et al., 2020a*; *Li et al., 2019*; *Perini et al., 2015*). The facilitation of pSTS activity may thus reflect encoding of social components of the CT-targeted gentle stroking touch in line with the reported effects of intranasal and oral OT on social cognition and attention (*Le et al., 2020*; *Le et al., 2021*; *Yao et al., 2018*; *Zhuang et al., 2021*; *Zhuang et al., 2022*). Interestingly, individuals with autism or higher autistic traits tend to find light social touch as the most aversive (*Ujiie and Takahashi, 2022*) and this corresponds to reduced responses in both the OFC and STS (*Kaiser et al., 2016*; *Voos et al., 2013*).

Plotting the time-course of responses in the mOFC and pSTS during gentle-stroking touch following OT treatment revealed that the main treatment effect was to prolong the neural response to the touch rather than the magnitude of the initial response. This may suggest that OT is acting to prolong the duration of the impact of the touch stimulation and possibly therefore facilitatory effects may have been even more pronounced for longer periods of touch. A classification analysis based on time courses revealed that responses in the three regions exhibiting significant changes during gentle-stroking touch (mOFC, mlOFC, STS) could significantly discriminate between both the intranasal and oral OT treatment groups and the PLC group (64–70% accuracy). Importantly however, the classification analysis did not find that activation in these three regions could discriminate between the intranasal and oral OT groups, further supporting the conclusion that the two routes of administration produced similar effects.

The finding in the current study that intranasal and oral administration of OT produced similar facilitation of behavioral and neural responses to gentle stroking touch provides further support for observations that some effects of exogenous treatment result from increased concentrations in the peripheral vascular system rather than following direct entry into the brain (*Yao et al., 2023*). One of the major considerations for studies choosing intranasal administration of OT has been from evidence that when administered by this route the peptide can gain direct access to the brain via the olfactory and trigeminal nerves (see *Quintana et al., 2021*; *Yao and Kendrick, 2022*). However, intranasal OT also increases concentrations of the peptide in peripheral blood, and accumulating evidence from both animal model and human studies support it having functional effects following administration via routes which only increase peripheral blood concentrations but do not permit direct entry into the brain (i.e. intravenous, intraperitoneal, subcutaneous and oral). Although, the intranasal administration route for OT produced significantly greater initial increases in plasma OT concentrations compared to an oral route in both the present study, and in two previous ones (*Kou et al., 2021*; *Xu et al., 2022*), this did not result in significant route-dependent differences, possibly due to the high doses involved. The broadly similar functional effects of intranasal and oral OT in the context of behavioral and neural responses to gentle stroking touch are also consistent with our previous findings for visual attention toward social and non-social stimuli (*Zhuang et al., 2022*; *Xu et al., 2022*), although there is also evidence for route-dependent effects of OT on neural and behavioral effects on responses to emotional faces (*Kou et al., 2021*).

The current study does not directly address the precise mechanism(s) whereby peripheral increases in OT concentrations facilitate neural and behavioral responses to gentle stroking touch via activation of CT fibers. However, despite original findings that only very small amounts of intravenously administered OT cross the BBB (*Mens et al., 1983*; *Kendrick et al., 1986*), the discovery that it can cross the BBB by binding to RAGE, and that RAGE knock out mice do not show brain and behavioral effects in response to peripherally administered OT (*Higashida et al., 2019*), has opened up a reconsideration of this route. Thus, observed neural and behavioral effects of OT could be via it entering the brain by crossing the BBB and acting on its receptors in the OFC and pSTS as well as in other associated regions (*Quintana et al., 2021*; *Quintana et al., 2019*). However, it is also possible that the effects of OT on neural and behavioral responses are mediated via stimulation of vagal afferents to the brain via receptors in the heart and gastrointestinal system (*Carter, 2014*; *Carter et al., 2020*). In agreement

with a previous study (*Triscoli et al., 2017*) we found that CT-targeted touch increased HRV (reflecting vagal activity) compared with rest. Vagal stimulation can also produce extensive activation in the brain, including in the frontal cortex and STS (*Chae et al., 2003*; *Iseger et al., 2020*). Studies have reported that OT can influence both parasympathetic and sympathetic nervous system activity (*Kemp et al., 2012*; *de Oliveira et al., 2012*), although in line with some other studies (*Martins et al., 2020b*; *Yao et al., 2023*) we did not observe any effects of intranasal or oral OT on SCR, heart rate or HRV during touch stimulation in the present study. Another possibility is that OT might be directly acting of CT fibers in the skin to increase their sensitivity to touch. In rats, OT has been found to exert analgesic effects by influencing OT receptors in cutaneous nociceptive fibers in the dorsal route ganglion (C- and A-δ but not A-β fibers; *González-Hernández et al., 2017*). Keratinocytes in the skin also express OT receptors (*Baumbauer et al., 2015*; *Deing et al., 2013*) and can modulate responses of sensory neurons to touch (*Talagas and Misery, 2019*; *Noguri et al., 2022*). However, in our current study we did not observe any indication that OT increased the sensitivity of CT fibers to touch given that only ratings of pleasantness were affected and not those of intensity. Behavioral and physiological indices of arousal in response to touch were also unaffected. Clearly, further experiments will be needed to help disentangle these different potential routes whereby OT is influencing CT fiber mediated social touch processing.

The question of why OT only influences the pleasure of CT fiber and not non-CT fiber-mediated touch can also not be resolved in the current study. The touch stimulation protocols used specifically aimed to differentially influence the cutaneous fiber types and subjects were informed that they would receive two types of massage from a professional which varied only in terms of pressure in order to minimize any possible differences in psychological or emotional responses. Indeed, both types of touch were perceived as pleasant and other potential psychological factors such as the perceived gender of the person giving the massage had no influence on the effects of OT. Since, CT fiber and non-CT fiber projections both to and within the brain are dissociable to some extent (*Case et al., 2021*; *Marshall et al., 2019*; *McGlone et al., 2014*) it is possible that the CT fiber projection system has more OT receptors. However, at this point, we cannot entirely rule out additional influences of OT on brain regions involved in processing other psychological or emotional responses associated with the experience of social touch.

The current study still has some limitations. Firstly, although using the well-established fNIRS technique made it easier to administer touch stimulation and take blood samples during brain activity recording it has inherent limitations in terms of being restricted to measurement of changes in the superficial layers of the cortex and being unable to measure activity in deeper key touch, salience and reward processing regions (*McGlone et al., 2012*; *Pinti et al., 2020*). Secondly, applying the gentle stroking touch and medium pressure massage onto subjects' legs for 15 min while they were sitting on a chair in a controlled manner is not as natural as in normal social interactions. Thirdly, we were unable to directly measure the actual pressure applied during the massage although intensity ratings were relatively similar across subjects and a professional masseur was used. Finally, it is possible that more frequent sampling of blood OT concentrations would have revealed increases during the touch and massage conditions.

In summary, our findings demonstrate that OT administered either intranasally or orally equivalently enhanced both pleasantness and medial OFC and pSTS responses to gentle stroking touch targeting CT fibers but not to medium pressure massage targeting non-CT fibers. Moreover, the facilitatory effects of OT on pleasantness ratings were significantly mediated by post-treatment changes in plasma OT concentrations. These findings support the conclusion that exogenously administered OT selectively facilitates neural and behavioral responses to stimulation of CT fibers during social touch via peripheral-mediated routes rather than by direct entry into the brain. Touch and massage interventions are increasingly demonstrated to have beneficial effects on brain development and function and it is likely that their effects on OT release and signaling play a major role in this (*Li et al., 2022*). Our findings demonstrating that OT treatment can facilitate behavioral and brain responses to gentle social touch/massage therefore may have important translational therapeutic implications for using OT-based interventions in individuals with aberrant responses to social touch such as in autism.

## Acknowledgements

This work was supported by the Key Technological Projects of Guangdong Province (grant number: 2018B030335001). We thank Weihua Zhao, Keshuang Li, and Xi Yang for their technical support.

## Additional information

### Funding

| Funder | Grant reference number | Author |
| --- | --- | --- |
| Key Technological Projects of Guangdong Province | 2018B030335001 | Keith M Kendrick |

The funders had no role in study design, data collection and interpretation, or the decision to submit the work for publication.

### Author contributions

Yuanshu Chen, Conceptualization, Data curation, Software, Formal analysis, Validation, Investigation, Visualization, Methodology, Writing - original draft, Project administration, Writing - review and editing; Haochen Zou, Chuimei Lan, Jing Wang, Yanan Qing, Wangjun Chen, Investigation; Xin Hou, Formal analysis, Visualization, Methodology; Shuxia Yao, Conceptualization, Supervision, Visualization, Methodology, Project administration, Writing - review and editing; Keith M Kendrick, Conceptualization, Resources, Supervision, Funding acquisition, Methodology, Writing - original draft, Project administration, Writing - review and editing

### Author ORCIDs

Yuanshu Chen ⬦ http://orcid.org/0000-0001-8500-7647
Keith M Kendrick ⬦ http://orcid.org/0000-0002-0371-5904

### Ethics

Clinical trial registration The Effects of Oxytocin Treatment on Social Touch; registration ID: NCT05265806; URL: https://clinicaltrials.gov/ct2/show/NCT05265806.
All subjects gave written informed consent prior to any study procedures. All experimental procedures were in accordance with the latest revision of the declaration of Helsinki and approved by the local ethics committee of the University of Electronic Science and Technology of China.

### Decision letter and Author response

Decision letter https://doi.org/10.7554/eLife.85847.sa1
Author response https://doi.org/10.7554/eLife.85847.sa2

## Additional files

### Supplementary files

• Supplementary file 1. Supplementary Tables 1A through 1D. Table 1A. Treatment effects on mood. Table 1B. Treatment effects on post-treatment plasma OT change (pg/ml). Table 1C. Confounding effects of individual perceived gender on behavioral pleasantness rating scores. Table 1D. Treatment effects on physiological indices.

• Supplementary file 2. CONSORT 2010 checklist.

• MDAR checklist

### Data availability

Individual data is plotted in Figure 2, 4 and 5. The group-level statistics are plotted in Figure 2, 3 and 6. The source data for Figure 2-6 of this study is available on Open Science Framework (https://osf.io/cykru/). The code for the condition decoding analysis (Figure 5 and Figure 5-figure supplement 1) was initially from *Emberson et al., 2017* (https://teammcpa.github.io/EmbersonZinszerMCPA/) and revised for group classification analysis.

The following dataset was generated:

| Author(s) | Year | Dataset title | Dataset URL | Database and Identifier |
|---|---|---|---|---|
| Chen Y, Kendrick K | 2023 | Effects of oxytocin on social touch | https://doi.org/10.17605/OSF.IO/CYKRU | Open Science Framework, 10.17605/OSF.IO/CYKRU |

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
