## [Editor Report]

The therapeutic promise of oxytocin to ameliorate deficiencies in social interactions and/or reward circuitry has been confounded by conflicting literature regarding routes of administration and regions of impact (i.e. central or peripheral). This important study systematically compares oral versus nasal administration on the pleasantness of gentle stroking, which is c-fiber mediated, and massage, which is multimodal. The convincing results are unambiguous that either route increases perceived pleasantness only to gentle stroking and that the effects, while perceived in the brain, are likely mediated peripherally.

---

## [Decision Letter]

**Decision letter after peer review:**

Thank you for submitting your article "Neural and behavioral responses to social touch targeting different cutaneous receptors after intranasal compared to oral oxytocin" for consideration by *eLife*. Your article has been reviewed by 3 peer reviewers, including Margaret M McCarthy as the Reviewing Editor and Reviewer #3, and the evaluation has been overseen by Floris de Lange as the Senior Editor. The following individuals involved in the review of your submission have agreed to reveal their identity: Monika Eckstein (Reviewer #1); Yi Huang (Reviewer #2).

Essential revisions:

1) In the procedure (and Figure 1a), it is not clear when blood samples were taken. This is especially interesting since the response is rather slow, so there might be a time delay after which a response in oxytocin levels might become evident after the specific touch and sum up over several samples. How many samples are taken in total, could you give them numbers? This might help understand the results on page 9 (describing them as "after treatment", "compared to baseline", "main effects" etc.). Adding time to the figure might also help.

2) Likewise, the order of the conditions is not clear to me. In Figure 1, there is only the massage condition mentioned. How many blocks were conducted?

3) The oral administration of oxytocin is very interesting as it is not very common. Please describe how it was performed. The oral versus intranasal application cannot be blinded, the only difference between them is towards higher peripherical levels after intranasal application. Maybe the authors can comment on how this relates to the main conclusion on which routes oxytocin acts.

4) The medium-pressure massage seems to target a mix of fibers, also including the CT touch. There is one study with pressure only that might complement the distinction:

Case, L. K., Liljencrantz, J., McCall, M. V., Bradson, M., Necaise, A., Tubbs, J., … and Bushnell, M. C. (2021). Pleasant deep pressure: expanding the social touch hypothesis. Neuroscience, 464, 3-11.

5) The authors didn't explain clearly why is it important to study the OT effect on different kinds of touch (CT vs. non-CT touch).

6) The authors mentioned that medium-pressure massage is different from CT in terms of targeting fibers. Simulation of the CT system activates brain regions involving reward and social-emotional processing. It is unclear what are the neural circuits of non-CT simulation. In the manuscript, the hypothesis related to the OT effect on non-CT touch is missing.

7) The authors claimed that intranasal OT can enter the brain and can also enter the peripheral circulation. However, it is unclear what is the mechanism of oral OT administration. The authors hypothesized that oral OT would have a similar effect to intranasal OT. It is unclear to me why oral and intranasal OT administration can be used to answer the question "whether the reported effects of exogenously administered OT on responses to social touch are mediated via direct entry into the brain or indirectly via increased concentrations in the peripheral circulation". How to distinguish the two mechanisms? The authors need to elaborate on this point.

8) The authors didn't explain the reasons for using fNIRS to investigate the neural mechanisms, any specific benefits? Why not fMRI, for example?

9) The introduction is a bit superficial. The objectives mainly focus on how the current study may fulfil the research gap. But it doesn't highlight the significance and implications.

10) It would be good to include a brief summary at the end of each result section.

11) The authors can discuss more the possible explanation for the results that OT selectively increases the hedonic pleasure of touch primarily stimulating CT-fibers rather than non-CT fiber pressure mechanoreceptors. As we know that OT plays a key role in social information processing. Would it be possible that gentle-stroking touch contains more social elements, such as satisfying emotional needs, as a kind of comfort? The pleasure elicited from medium-pressure massage touch is more related to the physical sensation.

12) It is unclear to me how this conclusion is drawn from the results – OT effect on C-touch via a peripherally-mediated route rather than direct entry into the brain. Why the possibility of entering the brain can be ruled out, given the authors reported no difference between intranasal and oral OT, and intranasal OT can enter the brain? More discussion would be helpful to understand the exact mechanism.

13) It would be good the have more elaboration on this point – "Our findings have important translational therapeutic implications for OT-based interventions in individuals with aberrant responses to social touch such as in autism".

14) My only suggestion for the authors is that they do themselves a disservice by not being more declarative of their conclusions in the title, which in its current form leaves the reader wondering what they found. I would suggest something more along the lines of: "Oral and nasal oxytocin equally enhance pleasantness of social touch by peripheral activation of c-fibers" or something similar.

15) Likewise, the authors' extensive imaging work is not mentioned anywhere in the Abstract but is an important component of the study. Many readers never go further than the Abstract and so including a reference to the findings is suggested.

*Reviewer #1 :*

The authors report on an experimental study with 4 treatment groups: intranasal oxytocin, oral oxytocin, intranasal placebo, and oral placebo. The placebo groups are pooled into one group. As within-subject manipulations, there were two conditions of social touch, either CT-targeted stroking or medium-pressure massage to the leg. Before and after the touch, blood samples were taken to assess peripherical oxytocin, fNIRS, as well as HRV and SCRs assessments, which were conducted parallel to the touch conditions. Subjective ratings were conducted after the touch conditions.

As study results, they report and present several main effects of oxytocin towards both kinds of touch and some specific effects for CT-touch.

They find increased subjective pleasantness of CT touch in both oxytocin groups as compared to placebo, although the pressure massage was perceived as more pleasant in general. The neural response in the OFC and STS (preselected ROIs, from which signal was assessed) is also increased in both verum groups as compared to placebo, especially in the CT-stroking condition. A classification analysis showed that neural patterns can be classified by treatment groups. Both touch conditions lead to an increase in peripherical oxytocin. A mediation model results in a connection between endogenous oxytocin levels that rose due to the treatment with self-reported pleasantness of CT-touch, – however no association with neural responses.

The authors conclude, that the effect of oxytocin is specific for CT-touch and is rather mediated via peripherical routes than via central routes.

A strength of the study is the design that allows the comparison of different possible mechanisms. This helps to add to the current important debate on oxytocin effects. The combination of measures gives a very detailed picture of the effects, although the neuroimaging data is limited to the preselected surface areas of assessment.

*Reviewer #2 :*

The objectives of the study are to examine whether (1) oxytocin (OT) directly enters into the brain and increases peripheral concentrations, and only peripheral pathway increases affect social touch, and (2) what kind of social touch, either targeting C-touch or other- (medium-pressure massage) types of cutaneous receptors would be affected. To achieve these goals, the authors used a randomized controlled trial. The research method includes behavioral measures and functional near-infrared spectroscopy (fNIRS) measures. The authors also did manipulation checks using physiological measures of arousal and measured the OT concentration changes in the blood. Taken together, the research methods are appropriate and comprehensive to address the research questions. At the behavioral and neural levels, the results consistently showed that exogenously administered oxytocin primarily modulates C-touch fiber-mediated social touch via a peripherally mediated route. The exact underlying mechanism of the OT effect on social touch still needs further investigation. The authors suggest that the findings may have potential therapeutic implications for OT-based treatment in individuals with aberrant responses to social touch such as in autism.

*Reviewer #3 :*

An extensive and conflicting literature on the mechanisms of oxytocin action in the context of social behavior and reward motivated the authors to conduct a systematic comparison of oral versus nasal administration of oxytocin in order to distinguish between central versus peripheral sites of action on the pleasantness of gentle touch versus massage.

The strengths of the study include:

1) Well-powered sample size and intra-subject design.

2) Systematic comparison of 2 routes of administration and 2 forms of touch, distinguished by c-fiber versus non-c-fiber involvement.

3) Measures of peripheral oxytocin levels both before and after the touch.

4) Measures of brain (cortical) activity via oxy-Hb across time before and after oxytocin and touch.

5) Careful attention to controls and confounding variables.

6) Limitations of the approaches appropriately acknowledged.

The authors achieved their goal by determining that both oral and nasal oxytocin increased the pleasure associated with a gentle touch by enhancing c-fiber responses in the periphery. There was no effect on massage.

The importance of the work is the practical implications for the use of oxytocin as a therapeutic for conditions involving deficits in social interactions or aversion to touch.

---

## [Author Response]

Essential revisions:1) In the procedure (and Figure 1a), it is not clear when blood samples were taken. This is especially interesting since the response is rather slow, so there might be a time delay after which a response in oxytocin levels might become evident after the specific touch and sum up over several samples. How many samples are taken in total, could you give them numbers? This might help understand the results on page 9 (describing them as "after treatment", "compared to baseline", "main effects" etc.). Adding time to the figure might also help.

We thank the reviewer for this comment and fully agree that more information on collection time and total numbers of blood samples should be provided. A total of 4 blood samples (6ml of each) were taken: before (Sample 1: baseline) and 30 minutes after treatment (Sample 2: post-treatment) as well as immediately after each session of touch stimulations (Sample 3 and 4: post-gentle touch/pressure massage). Condition-order was counterbalanced, so for example, blood sample 3 was immediately taken after the gentle touch for those who first received the applied gentle stroking touch around 50 minutes post-treatment and the final sample 4 after the medium pressure massage was taken no less than 85 minutes post-treatment. The same sampling time was followed for the blood sample 3 (post-pressure massage) and final sample 4 (post-gentle touch) for the other half of subjects who were delivered the pressure massage first and then gentle touch second. We have added these detailed descriptions of sampling time and total sample numbers in the “Blood samples and OT assay” section (page 10, lines 178-180). Meanwhile, we have also added more detailed information in Figure 1 and its legend (see Figure 1A on page 14).

2) Likewise, the order of the conditions is not clear to me. In Figure 1, there is only the massage condition mentioned. How many blocks were conducted?

We have added the corresponding information in the “Experimental procedure” section (see page 13, lines 253-256) and improved the figure to clearly show the specific design of the paradigm (see Figure 1A, page 14). As mentioned in the response to comment 1, condition-order was counterbalanced, that is, half of subjects were applied the gentle stroking touch followed by the pressure massage and the order was reversed for the other half. The touch paradigm consisted of two sessions and each session comprised 20 blocks of gentle stroking touch or medium pressure massage.

3) The oral administration of oxytocin is very interesting as it is not very common. Please describe how it was performed. The oral versus intranasal application cannot be blinded, the only difference between them is towards higher peripherical levels after intranasal application. Maybe the authors can comment on how this relates to the main conclusion on which routes oxytocin acts.

Following the reviewer’s suggestions, we have included the detailed description of intranasal and oral (lingual) OT administration (pages 11-12, lines 214-222). In short, for intranasal OT administration 3 puffs (0.1 ml for 1 puff) were alternatively applied to each nostril with 30 s between each puff, and for the oral OT administration, 6 puffs (also 0.1 ml for each puff) were applied onto the tongue (i.e. lingual spray) interspaced by 30 s and subjects were required not to swallow until just before the next puff was applied to allow time for absorption by oral blood vessels.

In response to the reviewer’s comments on the blinding issue of oral versus intranasal administration, we have further clarified the corresponding information to address the concerns raised by the reviewer (see page 12, lines 222-231). Basically the treatment was double-blind so that both subjects and experimenters could not know whether OT or PLC was actually administered. The debrief also showed that subjects couldn’t guess better than chance whether they had received OT or PLC (χ^2^ = 0.43, *p* = 0.512). However, the route of administration could not be blinded, as the referee noted. Subjects were told they would either receive treatment as a nasal or a lingual spray at the time of recruitment and this was also notified in their written informed consent. However, in line with our previous findings that oral and intranasal PLC administration do not produce different functional effects either at neural or behavioral levels (Kou et al., 2021; Zhuang et al., 2022) we also found no significant differences in pleasantness ratings of the gentle stroking touch and medium pressure massage between the intranasal and oral PLC groups (*p*s > 0.110) thereby confirming that knowledge of the route of administration had no effects. To further control for the lack of blinding in the groups receiving different routes of administration, blood samples and behavioral and neural responses were initially analyzed by experimenters blind as to the identity of the treatment group.

4) The medium-pressure massage seems to target a mix of fibers, also including the CT touch. There is one study with pressure only that might complement the distinction:Case, L. K., Liljencrantz, J., McCall, M. V., Bradson, M., Necaise, A., Tubbs, J., … and Bushnell, M. C. (2021). Pleasant deep pressure: expanding the social touch hypothesis. Neuroscience, 464, 3-11.

We thank the reviewer for raising this. Conventional medium pressure massage does involve both stroking and pressure and will therefore stimulate both CT and non-CT fibers. The pressure massage in our current study was however delivered without stroking in order to reduce CT fiber stimulation. We have now made this clearer throughout the text and specifically on page 12, lines 239-241. However, following the reviewer’s suggestion we have incorporated reference to the Case et al. study (page 4, lines 60-65; page 34, lines 643-645).

5) The authors didn't explain clearly why is it important to study the OT effect on different kinds of touch (CT vs. non-CT touch).

We have added more information in the introduction to clarify the rationale for why we considered comparing the effects of OT on different kinds of touch (pages 5-6, lines 74-92). Previous studies showing close association between OT and social touch and modulatory effects of intranasal OT on touch processing at both behavioral and neural levels, mostly employed CT-targeted touch such as the gentle stroking touch (Kreuder et al., 2017; Scheele et al., 2014) and medium pressure massage (also combined with stroking, see Chen et al., 2020b). Intranasal OT also enhances the pleasantness of gentle stroking touch administered indirectly via different materials independent of valence (Chen et al., 2020a). These findings consistently suggest that intranasal OT can enhance the hedonic properties of social touch. It is less clear whether it does the same for pleasant stimulation of non-CT fiber mechanoreceptors which can also be stimulated by social behaviors such as hugging and during medium pressure massage. Foot massage, for example, incorporates both gentle stroking and medium pressure massage and thus the observed release of OT (Li et al., 2019) and facilitatory effects of intranasal OT (Chen et al., 2020b) may be contributed by both CT and non-CT fibers. Moreover, there is some evidence that receiving frequent hugs can increase OT concentrations (Grewen et al., 2005; Light et al., 2005) although hugging is often accompanied by gentle stroking of the back or neck and so once again both CT and non-CT fibers may be involved. By identifying which types of social touch oxytocin influences, the present study could investigate which types of cutaneous afferent fibers are primarily involved in the facilitatory effects of OT during social touch processing.

6) The authors mentioned that medium-pressure massage is different from CT in terms of targeting fibers. Simulation of the CT system activates brain regions involving reward and social-emotional processing. It is unclear what are the neural circuits of non-CT simulation. In the manuscript, the hypothesis related to the OT effect on non-CT touch is missing.

We have now included some more information in the introduction concerning differences in neural responses to CT and non-CT fiber touch (page 4, lines 60-65):

“… Medium pressure touch in the form of hugging or massage can also be perceived as pleasant but mainly influences pressure receptors of non-CT fibers (Case et al., 2021; Field, 2010) and primarily targets the somatosensory cortex via the spinothalamic tract (McGlone et al., 2014). While neural substrates of pleasurable gentle stroking and medium pressure touch overlap to some extent they may also involve different parts of the somatosensory cortex and insula (Case et al., 2021).”

We have also added our specific hypothesis on the effect of OT on non-CT targeted medium pressure massage (page 8, lines 147-152):

“We hypothesized firstly that while subjects would find both the gentle stroking touch and medium pressure massage pleasurable, intranasal OT would particularly enhance behavioral and brain reward (OFC) and social processing (STS) region responses to gentle stroking touch targeting the CT fiber system in line with our previous studies (Chen et al., 2020a; Scheele et al., 2014) and not in response to the medium pressure massage targeting non-CT fibers.”

7) The authors claimed that intranasal OT can enter the brain and can also enter the peripheral circulation. However, it is unclear what is the mechanism of oral OT administration. The authors hypothesized that oral OT would have a similar effect to intranasal OT. It is unclear to me why oral and intranasal OT administration can be used to answer the question "whether the reported effects of exogenously administered OT on responses to social touch are mediated via direct entry into the brain or indirectly via increased concentrations in the peripheral circulation". How to distinguish the two mechanisms? The authors need to elaborate on this point.

In response to the reviewer’s comment, the key point for this issue is that while intranasally administered OT can produce effects following direct entry into the brain (via the olfactory and trigeminal nerves) it may also do so via raising peripheral blood concentrations and either influencing the brain via acting on its peripheral receptors (notably those influencing the vagus or possibly the skin or dorsal route ganglion) or crossing the blood brain barrier after binding to RAGE. When OT is administered via an oral route it can’t directly enter the brain and can only increase peripheral blood concentrations (see Yao and Kendrick, 2022 for an overview). Thus, if intranasal and oral administration of OT have identical functional effects then they are most likely to be mediated via increased peripheral concentrations. As we admit, we could not directly address the question of whether increased peripheral OT concentrations are influencing behavioral and neural responses to gentle touch via stimulation of vagal or skin or other peripheral receptors or following crossing the blood brain barrier after binding to RAGE. However, the lack of OT effects on arousal (behavioral ratings, skin conductance response and heart rate and perceived intensity) make it unlikely that it is influencing skin sensitivity or sympathetic nervous system responses to gentle stroking touch and the lack of effect on heart rate variability also suggests that it is not influencing parasympathetic vagal responses either. Thus, current findings suggest that increased peripheral concentrations of oxytocin may influence behavioral and brain responses by binding to RAGE and crossing the blood brain barrier, although this would need further experiments to confirm.

In the revised version of draft we have endeavored to make the logic and interpretation of our experimental design and findings clearer (see Introduction, lines: 93-124; 131-135 and Discussion, lines: 587-635). We have also now added analysis of heart rate and the high frequency component of heart rate variability to reinforce the absence of any treatment effects on sympathetic and parasympathetic responses (pages 22-23, lines 432-441).

8) The authors didn't explain the reasons for using fNIRS to investigate the neural mechanisms, any specific benefits? Why not fMRI, for example?

We have now provided a more detailed justification for our use of fNIRS. fNIRS is now widely used in neuroimaging (Pinti et al., 2020) and has a higher temporal resolution than functional MRI (Quaresima and Ferrar, 2019). In the context of the current experiment fNIRS also has the advantage that subjects could be more comfortable during the touch sessions compared to an MRI scanner and could be catheterized for taking multiple blood samples during recordings for measurement of plasma OT concentrations. Although, fNIRS only measures activity changes in the superficial cortex it allows recordings to be made from three of the main touch processing regions, the OFC, STS and somatosensory cortex and has therefore often been used in studies investigating touch processing (Bennett et al., 2014; Cruciani et al., 2021; Li et al., 2019).

Based on the reviewer’s comments, we have added the corresponding information in the introduction to explain why we chose to employ the fNIRS (page 8, lines 139-145). Nevertheless, we still acknowledged the weakness of fNIRS techniques on spatial resolution and restriction to outer cortex as one of the limitations of the present study (pages 34-35, lines 648-652).

9) The introduction is a bit superficial. The objectives mainly focus on how the current study may fulfil the research gap. But it doesn't highlight the significance and implications.

We thank the reviewer for pointing this out and have now included more emphasis on the potential implications of the research both in the introduction and in the discussion. Pleasurable social touch is of great importance for promoting social interactions and bonds. However, individuals with social dysfunction, such as in autism, often find touch unpleasant and can exhibit atypical affective and cortical reactivity (i.e. decreased OFC and STS responses) as well as altered tolerance of social/affective touch (Cascio et al., 2012; Green et al., 2015; Kaiser et al., 2016; Masson et al., 2020; Voos et al., 2013). Establishing the mechanisms whereby OT treatment can make touch more pleasurable may therefore help in developing its therapeutic use in conditions such as autism where social touch is often perceived as unpleasant.

Based on the reviewer’s comment in combination with comment 13 raised by other reviewers, we have added information about altered tolerance of touch in autism in the Abstract (page 2, lines 22-23) and Introduction (page 4, lines 48-50) and also expanded the discussion about therapeutic implications of OT in autism (page 31, lines 572-575; page 35, lines 665-671).

10) It would be good to include a brief summary at the end of each result section.

We have now included a brief summary at the end of each result section (see page 19, lines 371-374; page 21, lines 397-400; pages 22-23, lines 423-426, 437-441; page 27, lines 495-497 and page 28, lines 522-524).

11) The authors can discuss more the possible explanation for the results that OT selectively increases the hedonic pleasure of touch primarily stimulating CT-fibers rather than non-CT fiber pressure mechanoreceptors. As we know that OT plays a key role in social information processing. Would it be possible that gentle-stroking touch contains more social elements, such as satisfying emotional needs, as a kind of comfort? The pleasure elicited from medium-pressure massage touch is more related to the physical sensation.

We thank the reviewer for raising this point. We were careful to have both types of touch stimulation administered by a professional masseur and using a strategy where the manual manipulation of gentle stroking touch and medium pressure massage ensured the difference between the two types of social touch was only in terms of activating CT or non-CT afferent fibers and pressure. To help avoid possible different psychological and emotional responses to the two types of touch subjects were only informed that they would be given two different forms of massage which varied only in terms of pressure. Thus, we controlled as far as possible for subjects having a different psychological interpretation of the gentle stroking touch and medium pressure massage. Indeed, although both types of touch were perceived as pleasant the medium pressure massage was actually considered more pleasant, arousing and intense than the gentle stroking touch. The afferent projections from C-touch and non-C type fibers are dissociable to some extent at all levels of processing from the skin, spinal projections and brain circuitry and so differential responses to OT could reflect different OT receptor distributions in the two projection systems. Although we can’t rule out some contribution of psychological factors concerning how subjects interpreted the two different forms of touch we feel that at this point our findings are likely to reflect OT having a greater influence on CT fiber rather than non-CT fiber processing. We have now added another paragraph to the discussion considering why OT only influences the pleasantness and neural responses to gentle stroking touch as follows (page 34, lines 636-647):

“The question of why OT only influences the pleasure of CT fiber and not non-CT fiber mediated touch can also not be resolved in the current study. The touch stimulation protocols used specifically aimed to differentially influence the cutaneous fiber types and subjects were informed that they would receive two types of massage from a professional which varied only in terms of pressure in order to minimize any possible differences in psychological or emotional responses. Indeed, both types of touch were perceived as pleasant and other potential psychological factors such as the perceived gender of the person giving the massage had no influence on the effects of OT. Since, CT fiber and non-CT fiber projections both to and within the brain are dissociable to some extent (Case et al., 2021; Marshall et al., 2019; McGlone et al., 2014) it is possible that the CT fiber projection system has more OT receptors. However, at this point we cannot entirely rule out additional influences of OT on brain regions involved in processing other psychological or emotional responses associated with the experience of social touch.”

12) It is unclear to me how this conclusion is drawn from the results – OT effect on C-touch via a peripherally-mediated route rather than direct entry into the brain. Why the possibility of entering the brain can be ruled out, given the authors reported no difference between intranasal and oral OT, and intranasal OT can enter the brain? More discussion would be helpful to understand the exact mechanism.

Following the reviewer’s suggestions, we have now added a more detailed description of specific mechanisms whereby intranasal and oral OT may act and the rationale behind the study design in the introduction (pages 6-7, lines: 93-124; pages 7-8, lines 131-135) and have expanded the corresponding discussion (in pages 32-34, lines 587-635) (also see the response to comment 7). To re-iterate the response to comment 7, whereas intranasal OT can produce effects following either direct entry into the brain or through increasing peripheral concentrations, orally administered OT can only increase peripheral concentrations (and then enter the brain by crossing the blood brain barrier after binding to RAGE) and so if both routes of administration produce equivalent results then it must be the increases in peripheral concentrations that are of most importance.

13) It would be good the have more elaboration on this point – "Our findings have important translational therapeutic implications for OT-based interventions in individuals with aberrant responses to social touch such as in autism".

Following the reviewer’s suggestions, we firstly added corresponding information in the Abstract (page 2, lines 22-23) and Introduction (page 4, lines 48-50) and have also extended the corresponding discussion on the clinical application of OT in autism where individuals often find social touch unpleasant and exhibit atypical affective and cortical reactivity to social/affective touch (see page 31, lines 572-575 and page 35, lines 665-671) (also see the response to comment 9).

14) My only suggestion for the authors is that they do themselves a disservice by not being more declarative of their conclusions in the title, which in its current form leaves the reader wondering what they found. I would suggest something more along the lines of: "Oral and nasal oxytocin equally enhance pleasantness of social touch by peripheral activation of c-fibers" or something similar.

We thank the reviewer for raising the constructive suggestions on our title. We do agree with the reviewer that informing main findings in the title is quite helpful for readers to quickly have a general idea of the main results of a study. We did originally do this but were asked by the journal editorial office to change the format. Following the reviewer’s suggestions, we have now revised our title as:

“Oxytocin administration enhances pleasantness and neural responses to gentle stroking but not moderate pressure social touch by increasing peripheral concentrations”

15) Likewise, the authors' extensive imaging work is not mentioned anywhere in the Abstract but is an important component of the study. Many readers never go further than the Abstract and so including a reference to the findings is suggested.

We thank the reviewer for this suggestion although references in the abstract are usually not permitted and there are also word limitations. However, we have revised and lengthened the abstract and make the point that we are extending our previous findings. We also provide more details of the findings and their potential importance (page 2).

References

Baumbauer, K.M., DeBerry, J.J., Adelman, P.C., Miller. R.H., Hachisuka, J., Lee, K.H., Ross, S.E., Koerber, H.R., Davis, B.M., and Albers, K.M. (2015). Keratinocytes can modulate and directly initiate nociceptive responses. *ELife*. 4, e09674. https//doi.org/10.7554/*eLife*.09674.

Bennett, R. H., Bolling, D. Z., Anderson, L. C., Pelphrey, K. A., and Kaiser, M. D. (2014). FNIRS detects temporal lobe response to affective touch. Social Cognitive and Affective Neuroscience, 9(4), 470–476. https://doi.org/10.1093/scan/nst008

Cascio, C. J., Moana‐Filho, E. J., and Guest, S., et al. (2012). Perceptual and neural response to affective tactile texture stimulation in adults with autism spectrum disorders. Autism Research, 5(4), 231-244.

Case, L. K., Liljencrantz, J., McCall, M. V., Bradson, M., Necaise, A., Tubbs, J., … and Bushnell, M. C. (2021). Pleasant deep pressure: expanding the social touch hypothesis. Neuroscience, 464, 3-11.

Chen, Y., Becker, B., Zhang, Y., Cui, H., Du, J., Wernicke, J., Montag, C., Kendrick, K. M., and Yao, S. (2020a). Oxytocin increases the pleasantness of affective touch and orbitofrontal cortex activity independent of valence. European Neuropsychopharmacology: The Journal of the European College of Neuropsychopharmacology, 39, 99–110. https://doi.org/10.1016/j.euroneuro.2020.08.003

Chen, Y., Li, Q., Zhang, Q., Kou, J., Zhang, Y., Cui, H., Wernicke, J., Montag, C., Becker, B., Kendrick, K. M., and Yao, S. (2020b). The Effects of Intranasal Oxytocin on Neural and Behavioral Responses to Social Touch in the Form of Massage. Frontiers in Neuroscience, 14, 589878. https://doi.org/10.3389/fnins.2020.589878

Cruciani, G., Zanini, L., Russo, V., Mirabella, M., Palamoutsi, E.M., and Spitoni, G.F. (2021). Strengths and weaknesses of affective touch studies over the lifetime: A systematic review. Neuroscience and Biobehavoral Reviews, 127, 1-24. https://doi.org/10.1016/j.neubiorev.2021.04.012

Deing, V., Roggenkamp, D., Kühnl, J., Gruschka, A., Stäb, F., Wenck, H., … and Neufang, G. (2013). Oxytocin modulates proliferation and stress responses of human skin cells: implications for atopic dermatitis. Experimental dermatology, 22(6), 399-405.

de Oliveira, D. C. G., Zuardi, A. W., Graeff, F. G., Queiroz, R. H. C., and Crippa, J. A. S. (2012). Anxiolytic-like effect of oxytocin in the simulated public speaking test. Journal of Psychopharmacology, 26(4), 497–504. https://doi.org/10.1177/0269881111400642

Field, T. (2010). Touch for socioemotional and physical well-being: A review. Developmental Review, 30(4), 367–383. https://doi.org/10.1016/j.dr.2011.01.001

González-Hernández, A., Manzano-García, A., Martínez-Lorenzana, G., Tello-García, I. A., Carranza, M., Arámburo, C., and Condés-Lara, M. (2017). Peripheral oxytocin receptors inhibit the nociceptive input signal to spinal dorsal horn wide-dynamic-range neurons. Pain, 158(11), 2117–2128. https://doi.org/10.1097/j.pain.0000000000001024

Green, S. A., Hernandez, L., Tottenham, N., Krasileva, K., Bookheimer, S. Y., and Dapretto, M. (2015). Neurobiology of sensory overresponsivity in youth with autism spectrum disorders. JAMA psychiatry, 72(8), 778-786.

Grewen, K.M., Girdler, S.S., Amico, J., Light, K.C. (2005). Effects of partner support on resting oxytocin, cortisol, norepinephrine, and blood pressure before and after warm partner contact. Psychosomatic Medicine. 67(4), 531-8. https//doi.org/10.1097/01.psy.0000170341.88395.47

Kaiser, M. D., Yang, D. Y.-J., Voos, A. C., Bennett, R. H., Gordon, I., Pretzsch, C., Beam, D., Keifer, C., Eilbott, J., McGlone, F., and Pelphrey, K. A. (2016). Brain Mechanisms for Processing Affective (and Nonaffective) Touch Are Atypical in Autism. Cerebral Cortex (New York, N.Y.: 1991), 26(6), 2705–2714. https://doi.org/10.1093/cercor/bhv125

Kemp, A. H., Quintana, D. S., Kuhnert, R.-L., Griffiths, K., Hickie, I. B., and Guastella, A. J. (2012). Oxytocin increases heart rate variability in humans at rest: Implications for social approach-related motivation and capacity for social engagement. PloS One, 7(8), e44014. https://doi.org/10.1371/journal.pone.0044014

Kou, J., Lan, C., Zhang, Y., Wang, Q., Zhou, F., Zhao, Z., Montag, C., Yao, S., Becker, B., and Kendrick, K. M. (2021). In the nose or on the tongue? Contrasting motivational effects of oral and intranasal oxytocin on arousal and reward during social processing. Translational Psychiatry, 11(1), 94. https://doi.org/10.1038/s41398-021-01241-w

Kreuder, A.-K., Scheele, D., Wassermann, L., Wollseifer, M., Stoffel-Wagner, B., Lee, M. R., Hennig, J., Maier, W., and Hurlemann, R. (2017). How the brain codes intimacy: The neurobiological substrates of romantic touch. Human Brain Mapping, 38(9), 4525–4534. https://doi.org/10.1002/hbm.23679

Li, Q., Becker, B., Wernicke, J., Chen, Y., Zhang, Y., Li, R., Le, J., Kou, J., Zhao, W., and Kendrick, K. M. (2019). Foot massage evokes oxytocin release and activation of orbitofrontal cortex and superior temporal sulcus. Psychoneuroendocrinology, 101, 193–203. https://doi.org/10.1016/j.psyneuen.2018.11.016

Light, K. C., Grewen, K. M., and Amico, J. A. (2005). More frequent partner hugs and higher oxytocin levels are linked to lower blood pressure and heart rate in premenopausal women. Biological psychology, 69(1), 5-21.

Marshall, A.G., Sharma, M.L., Marley, K., Olausson, H., and McGlone, F.P. (2019). Spinal signalling of C-fiber mediated pleasant touch in humans. *ELife*. 8:e51642. https//doi.org/10.7554/*eLife*.51642.

Martins, D., Davies, C., De Micheli, A., Oliver, D., Krawczun-Rygmaczewska, A., Fusar-Poli, P., and Paloyelis, Y. (2020a). Intranasal oxytocin increases heart-rate variability in men at clinical high risk for psychosis: a proof-of-concept study. Translational Psychiatry, 10(1), 227.

Masson, H. L., de Beeck, H. O., and Boets, B. (2020). Reduced task-dependent modulation of functional network architecture for positive versus negative affective touch processing in autism spectrum disorders. NeuroImage, 219, 117009.

McGlone, F., Wessberg, J., and Olausson, H. (2014). Discriminative and affective touch: Sensing and feeling. Neuron, 82(4), 737–755. https://doi.org/10.1016/j.neuron.2014.05.001

Noguri, T., Hatakeyama, D., Kitahashi, T., Oka, K., and Ito, E. (2022). Profile of dorsal root ganglion neurons: Study of oxytocin expression. Molecular Brain, 15(1), 44.

Pinti, P., Tachtsidis, I., Hamilton, A., Hirsch, J., Aichelburg, C., Gilbert, S., and Burgess, P. W. (2020). The present and future use of functional near-infrared spectroscopy (fNIRS) for cognitive neuroscience. Annals of the New York Academy of Sciences, 1464(1), 5–29. https://doi.org/10.1111/nyas.13948

Quaresima, V., and Ferrari, M. (2019). Functional Near-Infrared Spectroscopy (fNIRS) for Assessing Cerebral Cortex Function During Human Behavior in Natural/Social Situations: A Concise Review. Organizational Research Methods, 22(1), 46–68. https://doi.org/10.1177/1094428116658959

Scheele, D., Kendrick, K. M., Khouri, C., Kretzer, E., Schläpfer, T. E., Stoffel-Wagner, B., Güntürkün, O., Maier, W., and Hurlemann, R. (2014). An oxytocin-induced facilitation of neural and emotional responses to social touch correlates inversely with autism traits. Neuropsychopharmacology, 39(9), 2078–2085. https://doi.org/10.1038/npp.2014.78

Talagas, M., and Misery, L. (2019). Role of Keratinocytes in Sensitive Skin. Frontiers in Medicine. 6, 108. https//doi.org/10.3389/fmed.2019.00108

Voos, A.C., Pelphrey, K.A., and Kaiser, M.D. (2013) Autistic traits are associated with diminished neural response to affective touch. Social Cognitive and Affective Neuroscience, 8(4), 378-86. https://doi.org/10.1093/scan/nss009.

Yao, S., and Kendrick, K. M. (2022). Effects of Intranasal Administration of Oxytocin and Vasopressin on Social Cognition and Potential Routes and Mechanisms of Action. Pharmaceutics, 14(2), 323. https://doi.org/10.3390/pharmaceutics14020323

Yao, S., Chen, Y., Zhuang, Q., Zhang, Y., Lan, C., Zhu, S., and Kendrick, K.M. (2023). Molecular Psychiatry, https://doi.org/1038/s41380-023-020-02075-2

Zhuang, Q., Zheng, X., Yao, S., Zhao, W., Becker, B., Xu, X., and Kendrick, K. M. (2022). Oral Administration of Oxytocin, Like Intranasal Administration, Decreases Top-Down Social Attention. The International Journal of Neuropsychopharmacology, 25(11), 912–923. https://doi.org/10.1093/ijnp/pyac059